# OTUD5 cooperates with TRIM25 in transcriptional regulation and tumor progression via deubiquitination activity

Fangzhou Li[1], Qianqian Sun[1], Kun Liu[1], Ling Zhang[2], Ning Lin[1], Kaiqiang You[3], Mingwei Liu[4], Ning Kon[5], Feng Tian [6], Zebin Mao[1], Tingting Li[3], Tanjun Tong[1], Jun Qin[4], Wei Gu[5], Dawei Li [2✉] & Wenhui Zhao [1✉]

Oncogenic processes exert their greatest effect by targeting regulators of cell proliferation. Studying the mechanism underlying growth augmentation is expected to improve clinical therapies. The ovarian tumor (OTU) subfamily deubiquitinases have been implicated in the regulation of critical cell-signaling cascades, but most OTUs functions remain to be investigated. Through an unbiased RNAi screen, knockdown of OTUD5 is shown to significantly accelerate cell growth. Further investigation reveals that OTUD5 depletion leads to the enhanced transcriptional activity of TRIM25 and the inhibited expression of PML by altering the ubiquitination level of TRIM25. Importantly, OTUD5 knockdown accelerates tumor growth in a nude *mouse* model. OTUD5 expression is markedly downregulated in tumor tissues. The reduced OTUD5 level is associated with an aggressive phenotype and a poor clinical outcome for cancers patients. Our findings reveal a mechanism whereby OTUD5 regulates gene transcription and suppresses tumorigenesis by deubiquitinating TRIM25, providing a potential target for oncotherapy.

[1] Department of Biochemistry and Biophysics, Beijing Key Laboratory of Protein Posttranslational Modifications and Cell Function, Peking University Health Science Center, 38 Xueyuan Road, 100191 Beijing, China. [2] Center for Translational Medicine, The Affiliated Zhangjiagang Hospital of Soochow University, 68 Jiyang West Road, 215600 Suzhou, China. [3] Department of Biomedical informatics, School of Basic Medical Sciences, Beijing Key Laboratory of Protein Post-translational Modifications and Cell Function, Peking University Health Science Center, 38 Xueyuan Road, 100191 Beijing, China. [4] State Key Laboratory of Proteomics, Beijing Proteome Research Center, Beijing, China. [5] Institute for Cancer Genetics, and Department of Pathology and Cell Biology, College of Physicians and Surgeons, Columbia University, 1130 St. Nicholas Avenue, New York, NY 10032, USA. [6] Department of Laboratory Animal Science, Peking University Health Science Center, 38 Xueyuan Road, 100191 Beijing, China. ✉email: daweili@suda.edu.cn; zhao6025729@bjmu.edu.cn

Protein modification by ubiquitin (Ub) found in all eukaryotic cells plays a pivotal regulatory role in numerous cellular processes[1–3]. This functional diversity is achieved by the ability of Ub to form topologically distinct signals, which can be reversed by the deubiquitinases (DUBs)[3–6]. Approximately 100 *human* DUBs have been identified and classified into six families: ubiquitin-specific proteases (USPs), ubiquitin carboxy-terminal hydrolases (UCHs), ovarian tumor proteases (OTUs), Machado-Joseph disease protein domain proteases (MJDs), JAMM/MPN domain-associated metallopeptidases (JAMMs), and the monocyte chemotactic protein-induced protein (MCPIP) family[7]. By regulating the ubiquitin system, a number of DUBs have emerged as alternative and important therapeutic targets for cancers[8].

The OTU subfamily of DUBs have been the focus of many studies and shown to function in numerous cellular processes[3]. For example, A20 functions as a central regulator of multiple nuclear factor κB (NF-κB)-activating signaling cascades[9–11]. Specifically, OTUD7B inhibits TRAF3 proteolysis to prevent aberrant noncanonical NF-κB activation by binding and deubiquitinating TRAF3[12]. It has been suggested that OTULIN cleaves Met1-linked polyubiquitin chains to dampen linear ubiquitin chain assembly complex (LUBAC)-mediated NF-κB signaling[13]. OTUD5, also called DUBA, has emerged as a critical regulator in multiple cellular processes, including DNA damage repair, transcription and innate immunity[14–18]. Our previous study indicated that OTUD5 promoted DNA double-strand break (DSB) repair by inhibiting Ku80 degradation[14]. OTUD5 has also been shown to regulate the DNA damage response by regulating FACT-dependent transcription at damaged chromatin[15]. In particular, OTUD5 participates in the negative regulation of IFN-I expression by downregulating the ubiquitination of TRAF3[18]. OTUD5 inhibits the production of IL-17A by blocking the UBR5-mediated proteasomal degradation of RORγt[17]. Moreover, OTUD5 interacts with PDCD5 in response to etoposides, which is a prerequisite for the stabilization and activation of p53[16,19]. However, OTUD5 functions in tumorigenesis have remained largely unknown to date.

Tripartite motif (TRIM) proteins constitute a subfamily of RING domain-containing proteins, including the E3 ubiquitin ligase family, which share a conserved N-terminal structure containing one RING domain, one or two zinc-finger domains named B-box(es) (B1 box or B2 box), and a coiled-coil region[20]. TRIMs have been implicated in a broad range of functions important to tumorigenesis because of the functions as E3 ubiquitin ligases and other non-E3 ubiquitin ligase activities[20,21]. One of the TRIM family members, TRIM25, is involved in a variety of pathways through which it participates in the regulation of cell proliferation and migration[22–27]. TRIM25 targets the negative cell cycle regulator 14-3-3σ for degradation and promotes cell proliferation[28]. TRIM25 also modulates the p53/MDM2 circuit, wherein TRIM25 deficiency increases p53 activity and p53-induced apoptosis[22,29,30]. TRIM25 has been shown to act as an oncogene by activating TGF-β pathways in *human* gastric cancer[25]. In addition, TRIM25 has been reported to be a global transcriptional regulator positioned at the center of breast cancer metastasis-related transcriptional networks. Depletion of TRIM25 drastically disrupts the expression of genes associated with metastasis[31]. Although accumulating evidence suggests TRIM25 roles in key pathways implicated in tumorigenesis, the exact mechanism by which TRIM25 modulates tumor progression remains unclear.

The tumor suppressor protein TRIM19, known as the promyelocytic leukemia protein (PML), forms large nuclear aggregates named PML nuclear bodies (PML-NBs). PML-NBs are present in almost every *human* cell type and appear as a macromolecular spherical structure[32–34]. PML function is frequently lost by reciprocal chromosomal translocation, which predisposes patients to acute promyelocytic leukemia (APL)[35]. PML-null *mice* are highly susceptible to tumor development when challenged by carcinogens, which highlights the crucial roles of PML in tumor suppression[35]. PML regulates the stability and transcriptional activity of the p53 tumor suppressor. PML-mediated p53 function was required to eradicate leukemia-initiating cells in a *mouse* model of APL[36]. PML also acts as a bona fide transcriptional target of p53 to potentiate its tumor suppressor effect, implicating PML in a positive feedback loop that controls p53 activity[37,38]. The prevalent understanding of PML induction is that interferons (IFNs) induce PML expression by initiating the JAK/STAT signaling pathway and IFN-stimulated response elements (ISREs) and IFNγ-activated sites (GAS) in the *PML* promoter[39,40]. IRF-8 was shown to be an obligatory regulator of IFNγ-induced PML expression[41]. Additionally, the stability of PML was shown to be regulated by Cullin3-KLHL20 ubiquitin ligase mediated ubiquitination[42]. Although PML is crucial for the activities of critical tumor-suppressive pathways, the mechanism by which PML is efficiently regulated remains unknown.

The general hallmark of cancer is uncontrolled cell proliferation. Targeting of the uncontrolled expansion and invasion can be exploited for the development of specific oncotherapies. In the current study, we find that OTUD5 knockdown accelerates cell growth in a TRIM25-dependent manner. Subsequently, OTUD5 is found to interact with and deubiquitinate TRIM25 in vitro and in vivo. Moreover, we demonstrate that OTUD5 plays a previously unknown role in transcriptional regulation and tumor suppression by deubiquitinating TRIM25, leading to the downregulation of PML and fewer PML-NBs. Finally, we assess OTUD5 roles in a *mouse* xenograft model and human cancers. Taken together, our study sheds light into regulation and function of the OTUD5-TRIM25 axis, suggesting it as a potential target for tumor therapy.

## Results

**Depletion of OTUD5 dramatically promotes cell proliferation.** In an attempt to identify the OTUs that regulate cell proliferation, we screened 14 *human* OTU DUBs through knockdown by short-hairpin RNA (shRNA) and determined their effects on cell proliferation using colony formation assays. The knockdown efficiency of the shRNAs was verified by real-time PCR (RT-PCR) (Supplementary Fig. 1A). The results of the colony formation assay showed that the knockdown of 8 of 14 OTU DUBs reduced the number of Hep3B cell colonies, whereas knockdown of 3 of 14 candidates promoted colony formation of the Hep3B cells. OTUD5-depleted cells formed nearly twice as many colonies as formed by the controls (Fig. 1a, b). To validate the results, two different small-interfering RNAs (siRNAs) were designed to target *human* OTUD5. Two different tumor cell lines, Huh7 and Hep3B, were then used to investigate whether the siRNA-mediated depletion of OTUD5 promoted cell proliferation. The messenger RNA (mRNA) and protein levels of OTUD5 in both the Huh7 and Hep3B cells were sufficiently knocked down using the siRNAs (Fig. 1c, d). The cells were then plated on the dishes and cultured for ten days to allow colonies to form. Consistently, there were significantly more colonies formed by the OTUD5-depleted cells than were formed by the control cells (Fig. 1e, f). Finally, all OTUD5-depleted Huh7 cells and Hep3B cells had exhibited higher cell proliferation rates than the controls (Fig. 1g, h).

**TRIM25 mediates OTUD5 depletion-induced accelerated-growth.** DUBs are involved in various pathways and signaling networks by pairing with E3 ubiquitin ligases in protein

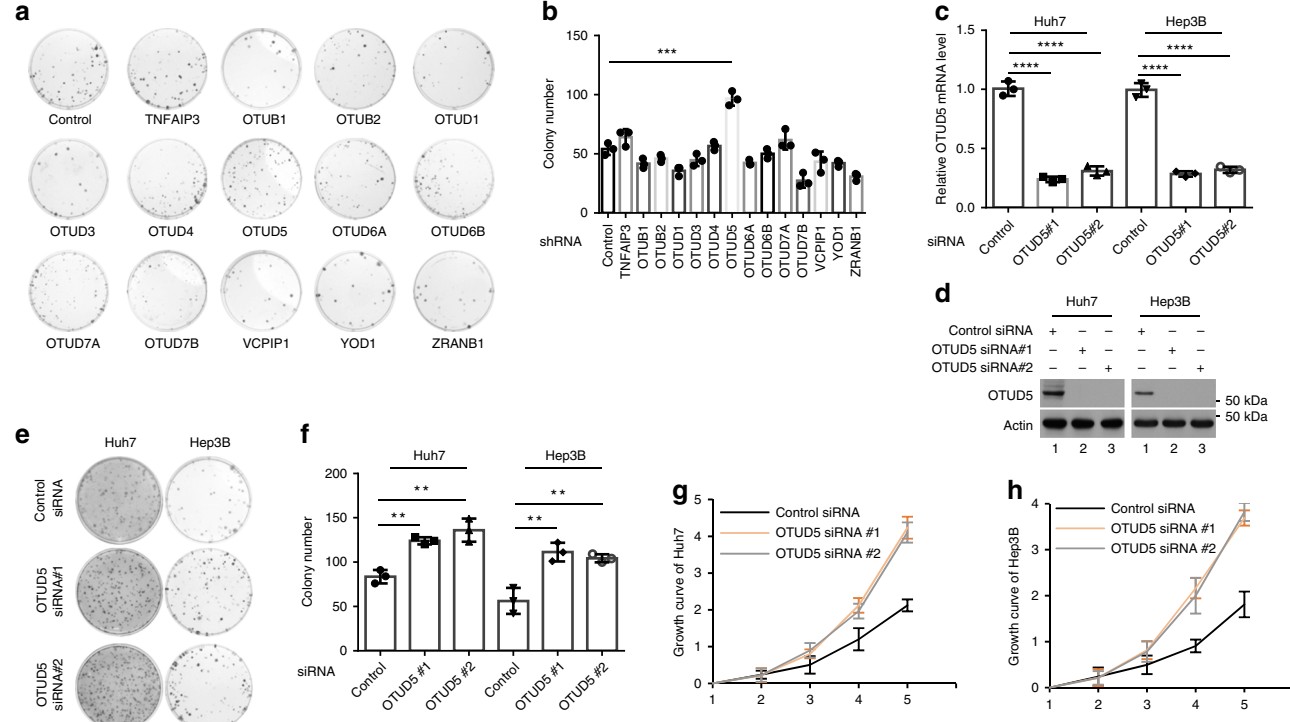

**Fig. 1 OTUD5 knockdown accelerated cell growth. a** The indicated DUBs were depleted by shRNA in Hep3B cells. DUB-depleted Hep3B cells were cultured for 10 days. The colonies were visualized by staining with methylene blue. **b** Quantitation of the colonies shown in **a**. Data are presented as the means ± SD of three biological experiments. ***$p = 0.0007$ (shControl versus shOTUD5), two-tailed unpaired $t$-test. **c** The mRNA level of OTUD5 in the OTUD5-depleted and negative control cell lines was determined by RT-PCR. GAPDH served as an endogenous control. Data are represented as means ± SD from three biological experiments. ****$p < 0.0001$ (siControl versus siOTUD5 #1 in the Huh7 cells), ****$p < 0.0001$ (siControl versus siOTUD5 #2 in the Huh7 cells), ****$p < 0.0001$ (siControl versus siOTUD5 #1 in the Hep3B cells), ****$p < 0.0001$ (siControl versus siOTUD5 #2 in the Hep3B cells), two-tailed unpaired $t$-test. **d** The protein expression of OTUD5 in the OTUD5-depleted and negative control cell lines was determined by western blot. β-actin served as a loading control. **e** Colony formation assay with the control cells or cells with OTUD5 depleted by one of the two different siRNAs. Cells were cultured for 10 days, transfected with siRNAs and stained by methylene blue. **f** Quantitation of the data shown in **e** from three repeated 35 mm dishes. Data are represented as means ± SD. **$p = 0.0012$ (siControl versus siOTUD5 #1 in the Huh7 cells), **$p = 0.0038$ (siControl versus siOTUD5 #2 in the Huh7 cells), **$p = 0.0060$ (siControl versus siOTUD5 #1 in the Hep3B cells), **$p = 0.0054$ (siControl versus siOTUD5 #2 in the Hep3B cells), two-tailed unpaired $t$-test. **g** and **h** Growth curves of the OTUD5-depleted Huh7 cells (**g**) and Hep3B cells (**h**) as determined by spectrophotometer. Data are presented as the means ± SD of three independent experiments with each performed in triplicate. Source data are provided as a Source Data file.

complexes[2,17,18,43–47]. In order to test whether OTUD5 associates with an E3 ubiquitin ligase to regulate cell proliferation, the stable Flag-OTUD5/Hep3B cell line was established by transfecting Hep3B cells with plasmids expressing pCIN4-Flag-OTUD5[48]. The cellular factors interacting with OTUD5 were purified by anti-Flag antibody-conjugated M2 beads. Mass spectrometry analysis revealed 2 unique peptides identical to the TRIM25 E3 ubiquitin ligase (Supplementary Data 1). The OTUD5 complex was then analyzed by western blotting, verifying the presence of TRIM25 (Fig. 2a). We confirmed the interaction between endogenous OTUD5 and TRIM25 by the immunoprecipitation (IP) of Hep3B cells lysates. TRIM25 was readily detected in the immunoprecipitates obtained with an anti-OTUD5 antibody but not the control IgG (Fig. 2b). Conversely, endogenous OTUD5 was immunoprecipitated with the TRIM25-specific antibody but not with a control antibody (Fig. 2c).

OTUD5 has an OTU domain in the central region of the polypeptide, and a ubiquitin interacting motif (UIM) at the conserved C-terminus (Fig. 2d). The OTUD5 UIM domain can bind both Lys[48]- and Lys[63]-linked polyubiquitin chains[15,18]. To determine the importance of UIM in the interaction between OTUD5 and TRIM25, we transfected Hep3B cells with plasmid constructs expressing Flag-tagged wild-type OTUD5 (Flag-OTUD5-WT) and Flag-tagged OTUD5 with the UIM domain

deleted (Flag-OTUD5-ΔUIM) (Fig. 2e, f). A significantly lower level of TRIM25 was obtained with Flag-OTUD5-ΔUIM than with Flag-OTUD5-WT, suggesting that the UIM domain is important for the interaction between OTUD5 and TRIM25. However, some TRIM25 coprecipitated with the Flag-OTUD5-ΔUIM, indicating that the UIM may not be the sole interaction site and that TRIM25 may bind to the other parts of the OTUD5 protein (Fig. 2f).

TRIM25 has been implicated in the increased proliferation of many cancer cell types, and knockdown of TRIM25 often results in attenuated tumor growth[26,28]. We therefore tested whether the increased colony formation after the knockdown of OTUD5 was mediated by TRIM25. First, a TRIM25-knockdown experiment was performed using two different siRNAs. The results showed that fewer colonies formed after TRIM25 was depleted in both the Huh7 and Hep3B cells (Fig. 2g, h). The efficiency of the TRIM25 knockdown by one of two different siRNAs was verified by western blot analysis (Fig. 2i). Second, we explored the role of TRIM25 in OTUD5-knockdown-induced increases of cell proliferation. Using lentivirus infection to express shRNA, we generated stable OTUD5 knockdown cells, into which the control siRNA or siRNA against TRIM25 was transfected. The OTUD5 depletion-induced increase in cell proliferation was partly abrogated by TRIM25 knockdown for both the Huh7 and Hep3B

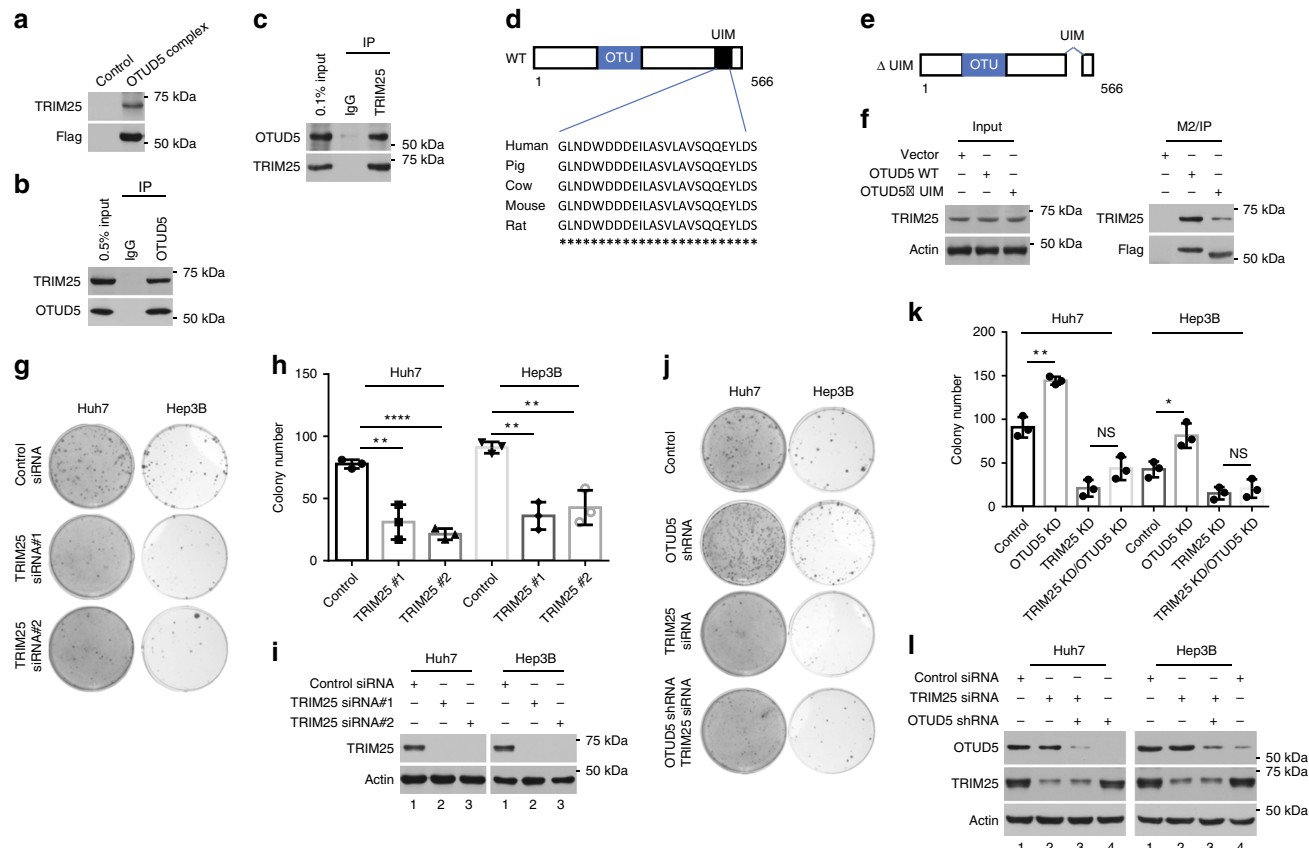

**Fig. 2 OTUD5 knockdown promoted cell growth in a TRIM25-dependent manner. a** TRIM25 was easily detected in the OTUD5 complex. **b** and **c** OTUD5 endogenously interacts with TRIM25. IP: immunoprecipitation. **d** Cross-species alignment of the UIM of OTUD5. WT, wild type. **e** Schematic representation of the UIM-deleted OTUD5 construct used in **f**. **f** UIM deletion inhibited the interaction between OTUD5 and TRIM25. **g** Huh7 and Hep3B cells were transfected with control siRNA or siRNA against TRIM25 (TRIM25 siRNA#1 and TRIM25 siRNA#2). The colonies were visualized by staining with methylene blue after the cells were cultured for 10 days. **h** Quantitation of colonies in **g**. Data represent the means ± SD from three repeated 35 mm dishes. **$p = 0.005$ (siControl versus siTRIM25 #1 in the Huh7 cells), ****$p < 0.0001$ (siControl versus siTRIM25 #2 in the Huh7 cells), **$p = 0.0013$ (siControl versus siTRIM25 #1 in the Hep3B cells), **$p = 0.0046$ (siControl versus siTRIM25 #2 in the Hep3B cells), two-tailed unpaired $t$-test. **i** TRIM25 knockdown efficiency by one of the two different siRNAs in both the Huh7 and Hep3B cells as determined by western blot analysis. **j** Representative images of the colonies showing that the growth acceleration of OTUD5 depletion was compromised by TRIM25 knockdown. **k** Quantitation of the data shown in (**j**). Data represent the means ± SD from three repeated 35 mm dishes. **$p = 0.0019$ (shControl versus shOTUD5 in the Huh7 cells), *$p = 0.016$ (shControl versus shOTUD5 in the Hep3B cells), two-tailed unpaired $t$-test. NS, non-significance. **l** The knockdown efficiency of OTUD5 and TRIM25 was assayed by western blot using an anti-OTUD5 antibody and anti-TRIM25 antibody. β-actin served as a loading control. Source data are provided as a Source Data file.

cells, suggesting that the cell proliferation-promoting effects of OTUD5 knockdown were possibly mediated by TRIM25 (Fig. 2j, k). The sufficient depletion of OTUD5 and TRIM25 by RNA interference in these cells was confirmed by western blotting (Fig. 2l).

**OTUD5 is critical for deubiquitinating TRIM25.** TRIM25 has been shown to exert its functions through its E3 ubiquitin ligase activity manifested through its RING domain. Indeed, smears corresponding to ubiquitinated TRIM25 were detected with an antibody against ubiquitin, a finding that was consistent with the autoubiquitination of E3 ubiquitin ligases (Fig. 3a). To test whether OTUD5 antagonizes the ubiquitination of TRIM25, we generated several constructs that produced wild-type OTUD5 protein, a catalytically inactive OTUD5 mutant protein (OTUD5/C224S), and a OTUD5 protein mutant lacking site-specific phosphorylation (OTUD5/S177A)[49–51]. In the in vitro deubiquitination assay, OTUD5 catalyzed the efficient deubiquitination of TRIM25, and the TRIM25 deubiquitination required OTUD5 DUB activity, as indicated by the catalytically inactive OTUD5

(OTUD5/C224S) failing to deubiquitinate TRIM25 (Fig. 3b, c, lane 2 versus lane 5). Moreover, phosphorylated OTUD5 was more efficient than wild-type OTUD5 in removing the ubiquitin chain of TRIM25 (Fig. 3b, c, lane 2 versus lane 3). Consistently, the deubiquitination activity of the phosphorylation-deficient mutant OTUD5 (OTUD5/S177A) was largely compromised, indicating that phosphorylation was critical for OTUD5 activity (Fig. 3b, c, lane 2 versus lane 4).

Furthermore, siRNA-mediated depletion of OTUD5 markedly increased TRIM25 polyubiquitination (Fig. 3d). Subsequently, when OTUD5 levels were reconstituted by transfection of wild-type OTUD5, the levels of ubiquitinated TRIM25 were reduced. In contrast, overexpression of the catalytic mutant OTUD5/C224S or the phosphorylation-deficient mutant OTUD5/S177A failed to reduce the levels of ubiquitinated TRIM25 in the OTUD5-knockdown cells (Fig. 3e). Finally, we transfected 293T cells with plasmids expressing Flag-TRIM25 and one of the three OTUD5 expression constructs and then immunoprecipitated the cell contents with M2 beads. In the absence of OTUD5, ubiquitinated TRIM25 could be purified by M2 beads. In contrast, the ubiquitination of TRIM25 was significantly

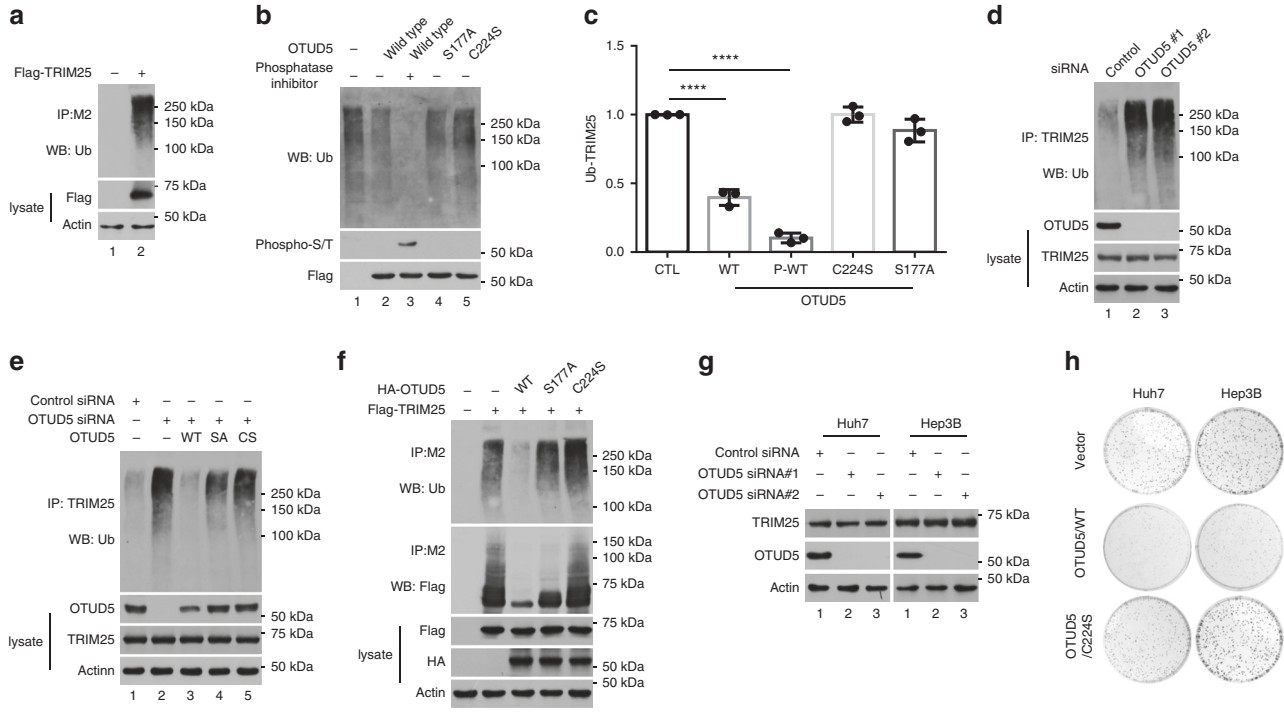

**Fig. 3 OTUD5 deubiquitinated TRIM25. a** TRIM25 was highly autoubiquitinated. **b** The in vitro deubiquitination assay using purified OTUD5 and TRIM25-ubs. OTUD5/WT, OTUD5/C224S or OTUD5/S177A was respectively incubated with TRIM25-ubs for the in vitro assay. **c** The in vitro deubiquitination assay performed in **b** was repeated three times. The signal intensity of the TRIM25-ubs was measured by Image-Lab software (Version 3.0 beta). Data are represented as means ± SD from three biological experiments. ****$p < 0.0001$ (CTL versus WT), ****$p < 0.0001$ (CTL versus p-WT), two-tailed unpaired $t$-test. p, phosphorylation. **d** The level of ubiquitinated TRIM25 increased following OTUD5 depletion by siRNA in the Hep3B cells. **e** Complementation with OTUD5/WT, but not with the catalytically inactive OTUD5 mutant, reduced the level of ubiquitinated TRIM25. **f** Co-overexpressing OTUD5/WT but not OTUD5/S177A or OTUD5/C224S reduced the level of ubiquitinated TRIM25. **g** Depletion of OTUD5 did not affect the level of TRIM25 protein. **h** The catalytic activity of OTUD5 was crucial for its function in the regulation of cell proliferation. Source data are provided as a Source Data file.

reduced in cells transfected with wild-type OTUD5 (Fig. 3f). The deubiquitination of TRIM25 was dependent on OTUD5 deubiquitinase activity and its activation by phosphorylation, as indicated by the cells coexpressing either OTUD5/C224S or OTUD5/S177A not having efficiently reduced levels of ubiquitinated TRIM25 (Fig. 3f). Additionally, co-overexpression of OTUD5 inhibited TRIM25 ubiquitination, phenocopying the TRIM25 K117R mutant, which was missing an important TRIM25 autoubiquitination site[52], implying that OTUD5 played a crucial role in TRIM25 autoubiquitination (Supplementary Fig. 2A). Interestingly, the stability of TRIM25 was not affected by the depletion of endogenous OTUD5, suggesting that TRIM25 was deubiquitinated by OTUD5 to modulate TRIM25 functions independent of TRIM25 stability (Fig. 3g). To determine which type of TRIM25 ubiquitin chains was affected by OTUD5, we purified lys48-UIM fusion proteins and lys63-UIM fusion proteins from bacteria for using in immunoprecipitation assays. The results of the western blotting for the IPed protein showed that the OTUD5 knockdown significantly increased lys63-ubiquitination of TRIM25 (Supplementary Fig. 3). Furthermore, we performed a colony formation assay with the catalytic activity mutant OTUD5. Overexpressed OTUD5/WT significantly inhibited colony formation, and OTUD5/C224S overexpression did not inhibit cell proliferation (Fig. 3h).

**TRIM25 knockdown promotes PML expression.** Containing a hallmark zinc-finger B-box DNA-binding domain (Fig. 4a), TRIM25 has been shown to function as a transcriptional suppressor in a signal network promoting breast cancer metastasis[31,53]. We hypothesized that TRIM25 regulates cell proliferation, at least in

part, by acting as a transcription factor. To explore the mechanisms by which TRIM25 contributes to tumor development, we performed an unbiased transcriptome analysis by RNA sequencing (RNA-seq, Supplementary Data 3–1). Differential gene expression was analyzed by supervised clustering focused particularly on critical cellular signaling cascades, such as those of the cell cycle, regulation of apoptosis and regulation of cell proliferation (Supplementary Fig. 1B). Differentially expressed genes are shown in Fig. 4b (log2-fold change > 1.5, $p < 0.05$). As shown in Fig. 4c, the PML tumor suppressor was among the differentially expressed genes showing the greatest variability. The elevated expression of PML in TRIM25-knockdown cells was corroborated by the increase in PML mRNA and protein levels (Fig. 4d, e).

PML is the major scaffold protein of the PML-NBs, which are macromolecular doughnut-shape structures, as revealed by electron microscopy[35,54]. To determine the impact of TRIM25 depletion on PML-NB formation, Huh7 and Hep3B cells were treated for 48 h with one of the two different siRNAs targeting TRIM25. After staining with the indicated antibody, PML-NBs were visualized by immunofluorescence using confocal microscopy. In the cells treated with control siRNA, very few PML-NB foci were observed. In contrast, the number of PML-NB foci was increased in the cells treated with TRIM25 siRNA, suggesting that TRIM25 depletion induced PML upregulation and promoted PML-NB formation (Fig. 4f, g). We then tested whether TRIM25 was critical for ubiquitinating PML. We found that co-overexpression with TRIM25 upregulated the ubiquitination level of PML (Supplementary Fig. 4A), consistent with a previous report demonstrating that TRIM25 plays roles through transcriptional and post-transcriptional mechanisms[31].

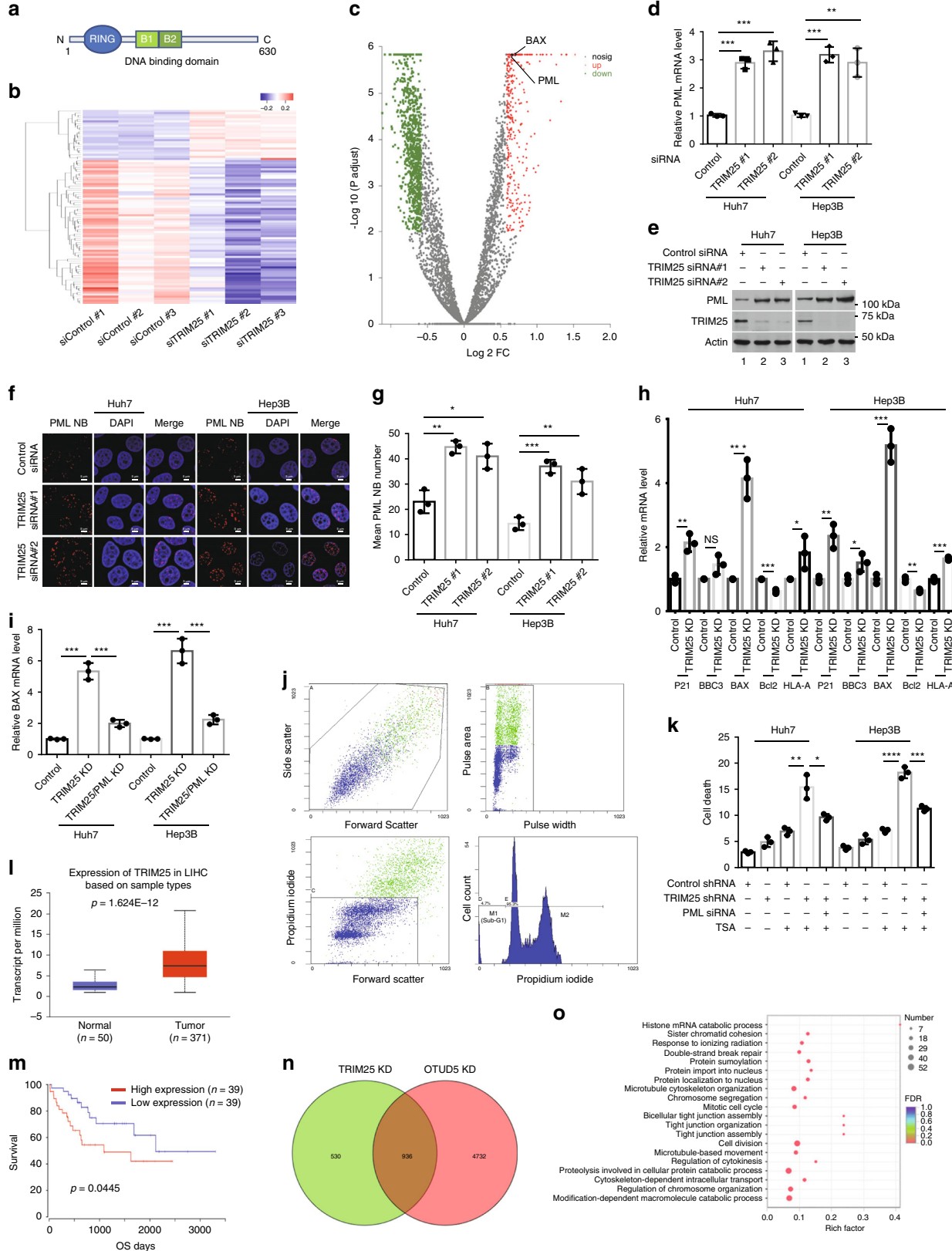

PML acts as a tumor suppressor by regulating the transcriptional functions of tumor suppressors such as p53, pRb, CBP and eIF4E[35,55]. To further establish the effects of the altered expression of PML in TRIM25-knockdown cells, the expression of PML target genes was assessed by RT-PCR. The relative expression levels of CDK1A, BBC3, BAX, and HLA-A were upregulated in the TRIM25-depleted cells, whereas the expression level of Bcl2 was only modestly reduced (Fig. 4h). Moreover, TRIM25 knockdown could no longer upregulated repression of BAX in PML-depleted cells (Fig. 4i). To further demonstrate the role of TRIM25 in the regulation of PML function, the effect of TRIM25 on Huh7 and Hep3B cell survival after trichostatin A

**Fig. 4 TRIM25 suppressed the expression of PML by targeting the TSS of *PML*. a** The TRIM25 structure is shown in a schematic diagram, B, B-box. **b** Heatmap of differentially expressed genes after TRIM25 knockdown. **c** PML represented one of the most differentially expressed genes following TRIM25 knockdown. **d, e** TRIM25 knockdown increased expression level of PML. Data are represented as means ± SD from three biological experiments. **$p < 0.01$, ***$p < 0.001$, two-tailed unpaired *t*-test. **f, g** TRIM25 depletion promoted PML-NBs formation. Scale bar, 5 μm. The average numbers were counted from three independent experiments. Data represent the means ± SD; $n = 50$ cells, data pooled across three experiments. *$p < 0.05$, **$p < 0.01$, ***$p < 0.001$, two-tailed unpaired *t*-test. **h** The effect of TRIM25 depletion on PML targets was determined by RT-PCR. Data are represented as means ± SD from three biological experiments. *$p < 0.05$, **$p < 0.01$, ***$p < 0.001$, two-tailed unpaired *t*-test. NS, non-significance. **i** RT-PCR was performed to test BAX expression in response to TRIM25/PML knockdown. Data are represented as means ± SD from three biological experiments. ***$p < 0.001$, two-tailed unpaired *t*-test. **j** Strategy to determine the population of su-G1 cells by DNA concentration based on PI staining. **k** The function of TRIM25 on cell survival regulation was mediated by PML. Data are represented as means ± SD from three biological experiments. *$p < 0.05$, **$p < 0.01$, ***$p < 0.001$, ****$p < 0.0001$, two-tailed unpaired *t*-test. **l** The expression of TRIM25 mRNA in normal *human* liver tissues (Normal, $n = 50$) and liver hepatocellular carcinoma tissues (LIHC, $n = 371$) was compared ($p < 1.6 \times 10^{-11}$). Box (25–75th percentiles) and whisker (minimum–maximum) plots for TRIM25 expression in controls and liver hepatocellular carcinoma patients; the horizontal line inside the box indicates the median (the 50th percentile). *p*-value calculated by Kruskal–Wallis test. $p = 1.624E-12$. The data were extracted from the TCGA LIHC database. **m** A lower level of TRIM25 was correlated with better overall survival in the LIHC patients OS, overall survival. $p = 0.0445$. **n, o** The genes overlapping in TRIM25 KD and OTUD5 KD were analysis by Gene Ontology (GO) analysis. Source data are provided as a Source Data file. KD knockdown.

(TSA) treatment was determined by flow cytometric analysis (BD FACSstation software version 6.0)[56,57]. As shown in Fig. 4k, treatment with TSA alone moderately induced cell death in the Huh7 and Hep3B cells (6.9% and 7.1% sub-G1), an effect that was significantly enhanced in the stable TRIM25-depleted cells (15.4% and 18.2% sub-G1). Notably, the knockdown of PML inhibited the cell death in TRIM25-depleted cells, in response to TSA treatment (Fig. 4k), suggesting that PML was possibly required for TRIM25-mediated function.

We then assessed TRIM25 expression through a *human* hepatocellular carcinoma (HCC) TCGA data set. The TRIM25 levels in the HCC tumor tissues were significantly higher than those in normal liver tissues (Fig. 4l). In addition, the high expression of TRIM25 was correlated with poor survival outcomes for HCC patients (Fig. 4m, $p < 0.05$). To evaluate the overlapping functions of OTUD5 and TRIM25, we performed RNA sequencing analysis with the OTUD5-depleted cells (Supplementary Data 3–2). A Gene Ontology (GO) analysis of the overlapping genes was performed. The enriched biological processes were dominated by cell cycle and cell division processes, implying that the OTUD5-TRIM25 axis may affect cell proliferation by regulating the cell cycle (Fig. 4n, o).

**TRIM25 autoubiquitination mediates transcription regulation.** To explore the mechanism related to the TRIM25-mediated regulation of PML expression, we performed chromatin immunoprecipitation (ChIP) qPCR analysis with the Huh7 and Hep3B cells (Fig. 5a). The results showed that TRIM25 mainly bound the transcription start site (TSS) region of the *PML* gene (2.6- and 3.1-fold, respectively; Fig. 5b), which is consistent with the notion that TRIM25 acts as a transcription suppressor by localizing to the promoter region of a target gene[31]. IRF-8 is critical for the IFN-gamma-induced expression of PML through the ISRE located within the promoter of *PML*[39,41]. ChIP-qPCR analysis was performed with the Huh7 and Hep3B cells transfected with plasmid expressing HA-tagged IRF-8. Indeed, a significant portion of recruited IRF-8 was detected by the HA Ab, not the IgG control. The recruitment of IRF-8 to the promoter of *PML* was significantly increased by the depletion of TRIM25 (Fig. 5c). These data suggested the possibility that TRIM25 repressed PML expression, at least partly, by inhibiting the recruitment of IRF-8 to the promoter of PML.

TRIM25 was originally described as an estrogen-responsive gene that was upregulated in estrogen receptor (ER)-positive mammary cells[58,59]. Although TRIM25 can potentially be regulated at the transcriptional level, we think that posttranslational modification of TRIM25 by the RING domain is involved,

at least in part, in the regulation for TRIM25. To explore this possibility, we constructed a plasmid expressing a mutant TRIM25 in which the RING domain was deleted (Fig. 5d, e). Consequently, the TRIM25 mutant lacking the domain (ΔRING), which is required for E3 ligase activity exhibited less ubiquitin modification (Fig. 5e). To test whether the RING domain was required for the enrichment of TRIM25 on the *PML* promoter, ChIP-qPCR analysis was performed with the Huh7 and Hep3B cells expressing HA-tagged TRIM25/WT or TRIM25/ΔRING. The results revealed that the recruitment of TRIM25 to the promoter was largely inhibited when the RING domain of TRIM25 was deleted (Fig. 5f). Consistently, the accumulation of TRIM25 K117R mutant at the *PML* TSS was reduced (Supplementary Fig. 2B). Expression of the TRIM25/WT but not TRIM25/ΔRING significantly reduced the PML expression levels in the TRIM25-depleted cells (Fig. 5g, h). Finally, we performed a colony formation assay to compare the effect of the TRIM25 K117R mutant and OTUD5 on cell proliferation. The results demonstrated that the TRIM25-KR mutant inhibited cell growth, consistently, cells with increased OTUD5 expression formed fewer colonies than the control cells (Supplementary Fig. 2C, D), implying that the autoubiquitination of TRIM25 is crucial for its function in cell proliferation regulation.

**OTUD5 regulates the transcriptional activity of TRIM25.** The RNA-seq analysis with OTUD5-depleted cells demonstrates that PML expression was downregulated upon OTUD5 knockdown (Supplementary Data 3–2) that OTUD5 binds to TRIM25 and functions as a TRIM25 deubiquitinase, which prompted us to assess whether OTUD5-mediated deubiquitination is involved in the regulation of TRIM25 transcriptional activity. The extent of TRIM25 recruitment to *PML* promoter was determined by ChIP-qPCR assay using OTUD5-depleted Huh7 and Hep3B cells and an antibody against TRIM25. Indeed, OTUD5 depletion increased the enrichment of TRIM25 on the promoter of *PML* (Fig. 6a).

Next, we assessed the impact of OTUD5 knockdown on PML expression. Total mRNA was isolated from the Huh7 and Hep3B cells transfected with control siRNA or siRNA against OTUD5 and used for RT-PCR analysis. The relative expression levels of PML were significantly reduced in response to OTUD5 depletion (Fig. 6b). Moreover, whole-cell extracts were prepared from control and OTUD5-depleted Huh7 and Hep3B cells. Western blot analysis demonstrated that the depletion of OTUD5 reduced the protein levels of PML in the Huh7 and Hep3B cells, a finding consistent with the RT-PCR results (Fig. 6c), suggesting that OTUD5 is required for the regulation of PML at the transcription level. To

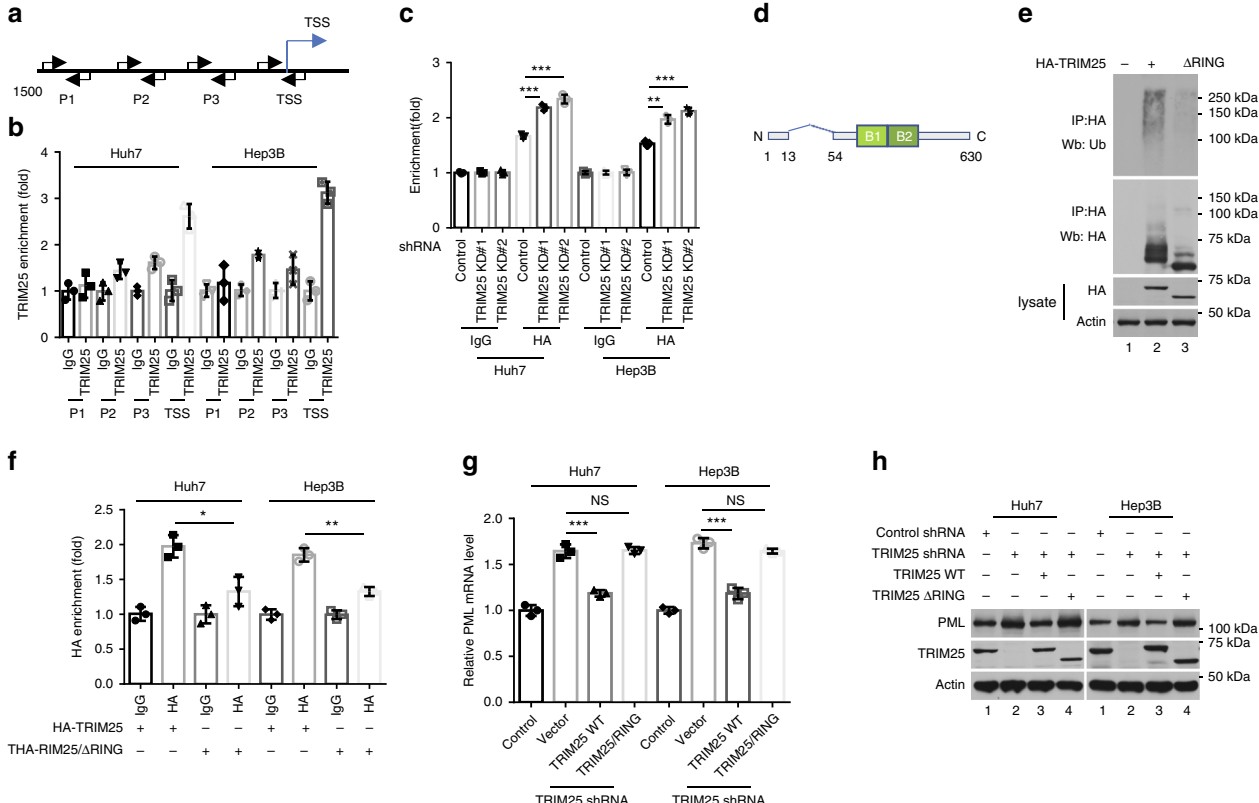

**Fig. 5 Transcriptional activity of TRIM25 was regulated by the RING domain. a** Schematic diagram of the ChIP-qPCR analysis of the *PML* promoter regions. P, promoter; TSS, transcription start site. **b** RT-PCR on ChIP samples was carried out at the indicated four regions (P1, P2, P3, and TSS). Fold enrichment represents the accumulation of TRIM25 compared with an IgG control on the target. Data represent the means ± SD of three independent experiments. **c** The recruitment of IRF-8 to the *PML* promoter was promoted after TRIM25 knockdown. ***$p = 0.0002$ (siControl versus siTRIM25 #1 in the Huh7 cells), ***$p = 0.0002$ (siControl versus siTRIM25 #2 in the Huh7 cells), **$p = 0.0010$ (siControl versus siTRIM25 #1 in the Hep3B cells), ***$p = 0.0002$ (siControl versus siTRIM25 #2 in the Hep3B cells), two-tailed unpaired *t*-test. **d** Schematic representation of the TRIM25 RING domain deletion construct used in **e**–**h**. **e** RING deletion inhibited ubiquitination of TRIM25. **f** The RING domain was crucial for the recruitment of TRIM25 to the target. Fold enrichment indicates the recruitment of HA-TRIM25 compared with that of an IgG control. Data show the means ± SD of three biological independent experiments. *$p = 0.0131$ (TRIM25 versus TRIM25/△RING in the Huh7 cells), **$p = 0.0014$ (TRIM25 versus TRIM25/△RING in the Hep3B cells), two-tailed unpaired *t*-test. **g**, **h** Complemental TRIM25 but not TRIM25/△RING suppressed PML expression in TRIM25 depleted cells. TRIM25 was depleted by a 3′-untranslated region (UTR)-targeting shRNA. Data represent the means ± SD; $n = 3$. ***$p = 0.0006$ (Vector versus TRIM25-WT in the Huh7 cells), ***$p = 0.0003$ (Vector versus TRIM25-WT in the Hep3B cells), two-tailed unpaired *t*-test. NS, non-significance. Source data are provided as a Source Data file.

exclude the possibility that OTUD5 regulated the abundance of PML by antagonizing the proteasome-mediated degradation of PML, an in vitro deubiquitination assay was performed. The results showed that OTUD5 did not cleave the ubiquitin chain of PML in vitro (Supplementary Fig. 4B). Furthermore, OTUD5 knockdown failed to affect the expression of PML in the stable TRIM25-depleted cells, indicating that the OTUD5 effect on PML expression was TRIM25-dependent. (Fig. 6d, e). Knocking down OTUD5 significantly inhibited the formation of PML-NBs, a finding that was consistent with the involvement of OTUD5 in the regulation of PML expression (Fig. 6f, g).

We further tested whether OTUD5 is involved in the regulation targets of TRIM25. We performed in vivo ubiquitination assays with two of the TRIM25 targets MAVS and SFN (14-3-3 sigma)[28,60,61]. Indeed, the overexpression of TRIM25 increased the ubiquitination level of MAVS and reduced the protein level of SFN. In addition, the ubiquitination level of MAVS was reduced following OTUD5 coexpression, and OTUD5 increased the protein level of SFN, demonstrating that OTUD5 was possibly involved in these signaling pathways to work with the TRIM25 E3 ubiquitin ligase in the protein complex (Supplementary Fig. 5).

**The OTUD5-TRIM25 axis regulates liver cancer development.** We tested whether the OTUD5-TRIM25 axis would affect tumor growth in a *mouse* model. Hep3B cells were used for a xenograft tumors growth assay in nude *mice*. Thirty 4- to 6-week-old male nude *mice* were allocated to six groups of five *mice* each. Each *mouse* received 10⁶ cells of the control-infected or gene-KD Hep3B cells (OTUD5 KD, TRIM25 KD, PML KD, OTUD5/TRIM25 KD, or TRIM25/PML KD) by subcutaneous injection in both flanks. On the 15th day after injection, all of the *mice* were sacrificed and the tumors were excised and weighed. As shown in Fig. 7a, b, the tumor spheres derived from the OTUD5-depleted cells and PML-depleted cells grew faster and had greater weight than those excised from the control cells, while TRIM25 knockdown inhibited tumor sphere formation. In addition, the OTUD5/TRIM25 double knockdown cells formed smaller tumor sphere than did the OTUD5-depleted cells, demonstrating that the OTUD5-TRIM25 axis plays an important role in the regulation of tumor growth.

The knockdown efficiency of OTUD5 and the levels of TRIM25 and PML in the tumor tissues formed by the OTUD5-depleted cells were determined by RT-PCR. The results revealed that OTUD5 was efficiently depleted. The mRNA levels of PML

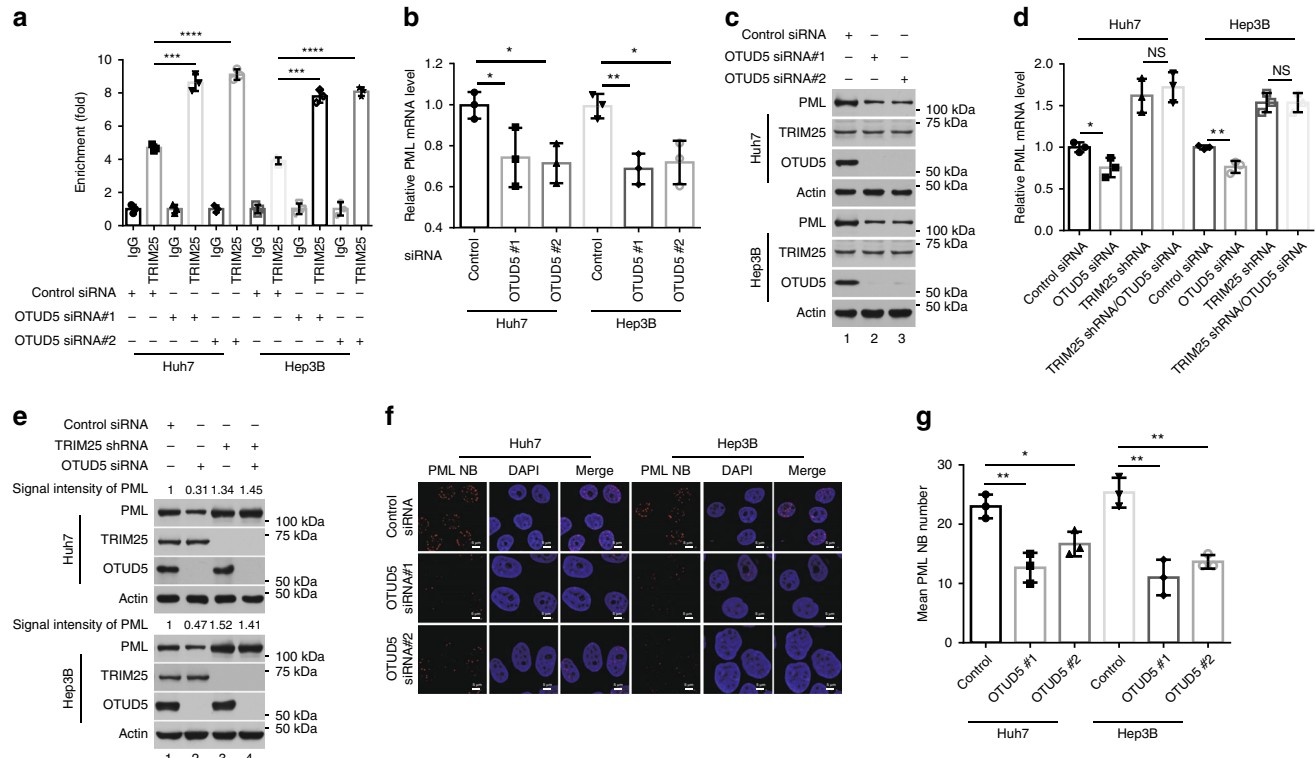

**Fig. 6 OTUD5 knockdown inhibited PML expression and PML-NBs formation. a** TRIM25 recruitment onto the *PML* promoter was increased in response to OTUD5 depletion. Data indicate the means ± SD, *n* = 3 for technical replicates. ***p = 0.0002 (siControl versus siOTUD5 #1 in the Huh7 cells), ****p < 0.0001 (siControl versus siOTUD5 #2 in the Huh7 cells), ***p = 0.0001 (siControl versus siOTUD5 #1 in the Hep3B cells), ****p < 0.0001 (siControl versus siOTUD5 #2 in the Hep3B cells), two-tailed unpaired *t*-test. **b**, **c** OTUD5 depletion inhibited the expression of PML. Data indicate the means ± SD from three biological replicates. *p = 0.0498 (siControl versus siOTUD5 #1 in the Huh7 cells), *p = 0.0138 (siControl versus siOTUD5 #2 in the Huh7 cells), **p = 0.0051 (siControl versus siOTUD5 #1 in the Hep3B cells), *p = 0.0173 (siControl versus siOTUD5 #2 in the Hep3B cells), two-tailed unpaired *t*-test. **d**, **e** OTUD5 knockdown inhibited PML expression in a TRIM25-dependent manner. **d** Data represent the means ± SD from three biological replicates. *p = 0.0320 (siControl versus siOTUD5 in the Huh7 cells), **p = 0.0058 (siControl versus siOTUD5 in the Hep3B cells), two-tailed unpaired *t*-test. NS, non-significance. **e** Western blotting was performed for the proteins from the whole-cell extract using the antibodies indicated. PML was quantified by Image-Lab software (Version 3.0 beta). **f** Depletion of OTUD5 reduced the formation of PML-NBs in the Huh7 and Hep3B cells. Representative images are shown. Scale bar, 5 μm. **g** The average number of PML-NB foci per cell was counted for three independent experiments and plotted as indicated. Data represent the means ± SD; *n* = 50 cells, data were pooled across three experiments. **p = 0.0051 (siControl versus siOTUD5 #1 in the Huh7 cells), *p = 0.0191 (siControl versus siOTUD5 #2 in the Huh7 cells), **p = 0.0032 (siControl versus siOTUD5 #1 in the Hep3B cells), **p = 0.0019 (siControl versus siOTUD5 #2 in the Hep3B cells), two-tailed unpaired *t*-test. NS, non-significance. Source data are provided as a Source Data file.

and BAX were downregulated upon OTUD5 knockdown (Fig. 7c). Consistently, the western blot analysis results demonstrated that OTUD5 was downregulated in the tumor spheres derived from the OTUD5-depleted cells. The protein level of PML was moderately reduced, whereas the level of TRIM25 was not altered by OTUD5 knockdown (Fig. 7d).

Finally, we assessed the OTUD5 expression and its possible clinical significance in *human* cancers. We examined OTUD5 expression in normal liver tissue and in primary liver cancer using a *human* liver tissue microarray (Cohort 1, ALENA Biotechnology, Xi'an, China; Supplementary Data 5). OTUD5-staining was observed to be distributed in both the nucleus and cytoplasm of the hepatocytes and tumor cells (Fig. 7e and Supplementary Fig. 6A, B). High OTUD5 staining (intensity greater than + +) was detected in 6 of the 10 normal liver tissues, accounting for 60.0%. However, high expression of OTUD5 was detected in only 28 of the 108 tissues from patients with primary liver cancer, accounting for 25.9%. OTUD5 expression was significantly downregulated in the primary liver cancer tissues compared with that in the normal liver tissues (Fig. 7e and Table 1, *p* < 0.05 by Fisher's exact test). Furthermore, the expression of OTUD5 was correlated with sex and tumor grade

in patients with primary liver cancer. High OTUD5 expression was detected in low-grade tumors, indicating that a high level of OTUD5 was correlated with a higher differentiation status. To further validate the OTUD5 expression in primary liver cancer, we assessed OTUD5 expression and its correlation with clinical features in another cohort of HCC patients (Cohort 2). OUTD5 expression was measured in both tumor and paired non-tumorous tissues from 90 HCC patients by IHC (Supplementary Data 6 and Supplementary Fig. 7). The results demonstrated that OTUD5 expression was downregulated in 38 (43.2%) patients, unchanged in 26 (29.5%) patients, and upregulated in 24 (27.3%) patients. OTUD5 levels were significantly downregulated in the HCC tissues compared with the noncancerous tissues (*p* < 0.05, Chi-square test; Fig. 7f). Importantly, OTUD5 expression was significantly correlated with tumor grade, tumor size and TNM stage (Table 2 and Fig. 7f), suggesting that lower OTUD5 expression correlates with more aggressive disease in HCC patients.

**OTUD5 displays general tumor-suppressive properties.** We then tested whether OTUD5 had general tumor-suppressive

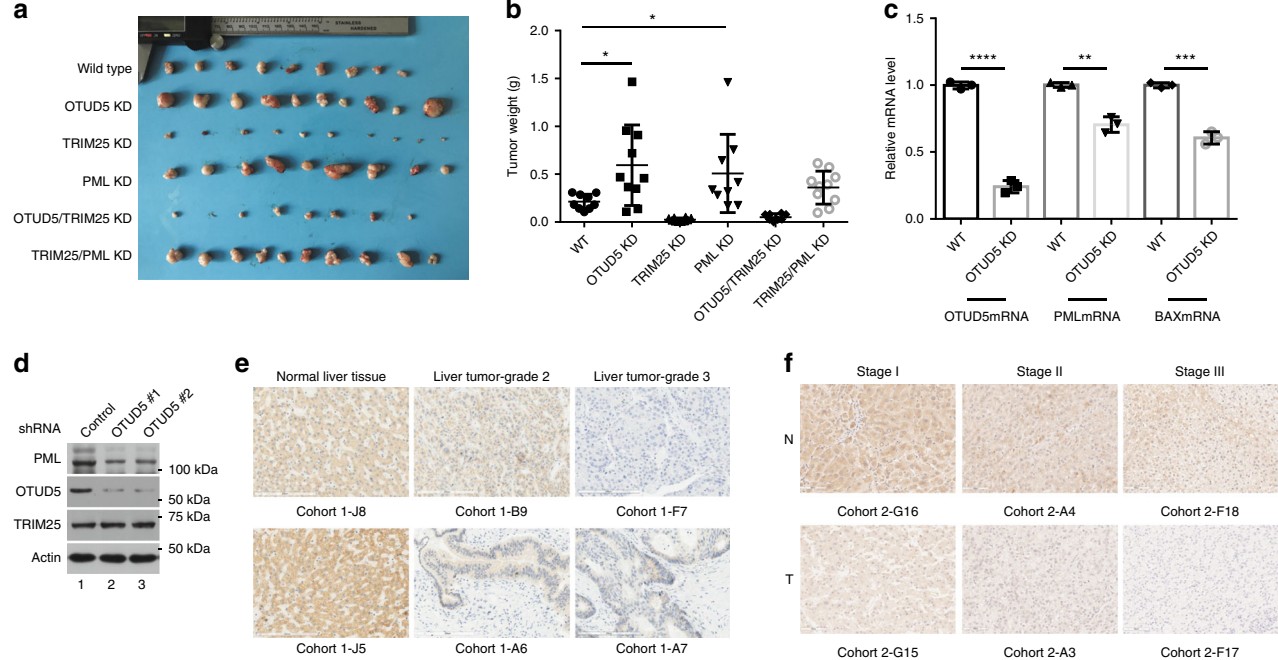

**Fig. 7 The OTUD5-TRIM25 axis regulated tumor sphere formation of Hep3B cells. a** Tumors derived from all groups are presented. **b** Tumor mass was measured in all groups and compared. Data are represented as means ± SD from biological repeats. WT ($n = 9$); OTUD5 KD ($n = 10$); TRIM25 KD ($n = 10$), PML KD ($n = 9$), OTUD5/TRIM25 KD ($n = 8$), TRIM25/PML KD ($n = 10$). **c** The mRNA of the OTUD5-depleted xenografted cells was extracted and prepared for RT-PCR. Data are represented as means ± SD from three biological experiments. ****$p < 0.0001$ (OTUD5 mRNA), **$p = 0.0011$ (PML mRNA), ***$p = 0.0002$ (BAX mRNA), two-tailed unpaired $t$-test. **d** The knockdown efficiency of OTUD5 and the expression levels of TRIM25 and PML from tumor tissue extracts were determined by western blot analysis. **e** The level of OTUD5 was downregulated and correlated with tumor grade of the primary liver cancer. Representative photographs of OTUD5 immunoreactivity in normal liver tissues (J8 and J5) and tumor tissues (B9 for grade 2 HCC and F7 for grade 3 HCC; A6 for grade 2 CCC; and A7 for grade 3 CCC) in a liver tissue microarray taken from Cohort 1 patients are shown (magnification 100×). HCC hepatocellular carcinoma, CCC cholangiocellular carcinoma. Scale bar, 200 μm. **f** The level of OTUD5 was downregulated and correlated with tumor TNM stage of the HCC samples. Representative photographs of OTUD5 immunoreactivity in paired noncancerous liver tissues (G16, A4, and F18) and HCC tissues (G15 for Stage I, A3 for Stage II, and F17 for Stage III) of a liver tissue microarray taken from Cohort 2 patients are shown (magnification 100×). HCC hepatocellular carcinoma. Scale bar, 100 μm. Source data are provided as a Source Data file.

properties. Two different lung tumor cell lines, H1299 and A549, were used to investigate whether siRNA-mediated depletion of OTUD5 also promoted cell proliferation. The results demonstrate that OTUD5 knockdown had a proliferation-promoting effect in both the H1299 cells and A549 cells (Fig. 8a). In addition, the protein level of PML was reduced after OTUD5 depletion (Fig. 8b). To further validate OTUD5 roles in tumor suppression, we investigated the OTUD5 expression and clinical significance in *human* non-small-cell lung carcinoma (NSCLC). The mRNA level of OTUD5 was markedly decreased in tissues from NSCLC tissues compared with that in the paired adjacent noncancerous tissues (Fig. 8c, left). Consistently, OTUD5 immunostaining was observed in normal lung and tumor cells, with higher staining in the control cells than in the cancer cells (Fig. 8d, left). To assess the OTUD5-mediated PML regulation in tumor development under more physio pathologically relevant conditions, we also investigated PML expression and its correlation with OTUD5 levels in NSCLC. The PML expression was markedly down-regulated in the tumor tissues than it was in the adjacent non-cancer tissues (Fig. 8c, right and 8d, right). PML staining was detected in both the cytoplasm and nucleus of the epithelial cells from tumor and nontumor tissues (Fig. 8d, right). Importantly, OTUD5 expression was significantly and positively correlated with PML levels in both tumor and nontumor tissues (Fig. 8e), supporting the supposition that OTUD5 may regulate PML in NSCLC patients. Furthermore, we assessed the correlation between OTUD5 expression, clinical pathological status and patient survival using the TCGA database. OTUD5 expression

was significantly correlated with tumor size, lymph node invasion and TNM stage in the NSCLC patients (Table 2). Additionally, patients with elevated OTUD5 expression had significantly longer overall survival compared to patients bearing tumors that expressed low levels of OTUD5 transcripts in NSCLC (lung adenocarcinoma, LUAD and lung squamous cell carcinoma, LUSC), PADD, CESC, LGG, and BLCA (Fig. 8f). Taken together, these findings indicated that OTUD5 displays strong tumor-suppressive properties with functional consequences in different tumor.

## Discussion

In this study, we discovered OTUD5 function in regulating cell proliferation and suppressing tumor, as demonstrated by the increased cell proliferation after RNAi-mediated OTUD5 knockdown and by the increased tumor growth after OTUD5 knockdown in a xenograft *mouse* model. Specifically, these unique functions of OTUD5 were mediated by TRIM25 and by the regulated PML expression through the interplay between the E3 ubiquitin ligase TRIM25 and its deubiquitinase OTUD5. Subsequently, we showed that TRIM25 blocked PML expression to promote cell proliferation and, potentially, tumorigenesis, which was prevented by OTUD5, suggesting that OTUD5 functions by regulating the TRIM25/PML axis. RING domain deleted and K117R mutants of TRIM25 had substantially inhibited functions, implying that OTUD5 functions by regulating the ubiquitination of TRIM25. Moreover, reduced expression of OTUD5 was associated with an aggressive tumor phenotype and

**Table 1 OTUD5 was correlated with clinical pathologic characteristics of liver cancer patients.**

| Variables | OTUD5 expression level (Cohort 1) | | | | OTUD5 expression level (Cohort 2) | | | |
|---|---|---|---|---|---|---|---|---|
| | Total cases (n = 108) | High | Low | p-value | Total cases (n = 89) | High | Low | p-value |
| Age | | | | | | | | |
| <50 | 47 | 11 | 36 | 0.662 | 27 | 12 | 15 | 0.446 |
| ≥50 | 61 | 17 | 44 | | 62 | 33 | 29 | |
| Sex | | | | | | | | |
| Male | 80 | 25 | 55 | 0.044* | 73 | 36 | 37 | 0.615 |
| Female | 28 | 3 | 25 | | 16 | 9 | 7 | |
| Tumor type | | | | | | | | |
| HCC | 93 | 27 | 66 | 0.109 | 89 | 45 | 44 | NA |
| CCC | 15 | 1 | 14 | | 0 | 0 | 0 | |
| Cirrhosis | | | | | | | | |
| Yes | NA | NA | NA | NA | 17 | 8 | 9 | 0.748 |
| No | NA | NA | NA | | 72 | 37 | 35 | |
| Tumor grade[a] | | | | | | | | |
| I–II | 55 | 19 | 36 | 0.049* | 14 | 11 | 3 | 0.039* |
| III | 51 | 9 | 42 | | 75 | 34 | 41 | |
| Tumor size | | | | | | | | |
| T1–T2 | NA | NA | NA | NA | 51 | 30 | 21 | 0.040* |
| T3–T4 | NA | NA | NA | | 31 | 11 | 20 | |
| TNM stage | | | | | | | | |
| I–II | NA | NA | NA | NA | 51 | 30 | 21 | 0.040* |
| III–IV | NA | NA | NA | | 31 | 11 | 20 | |

A total of 110 and 90 tumor samples in the tissue microarrays for primary liver cancer from Cohort 1 and Cohort 2, respectively, were immunostained. The OTUD5 expression level was stratified as high (scores ≥ 2) and low (scores < 2) expression. The staining for E7 was missing from Cohort 1. The staining for J1 was excluded from the statistics because of mixed hepatocellular carcinoma and normal live tissue in the Cohort 1 samples.
*HCC* hepatocellular carcinoma, *CCC* cholangiocellular carcinoma.
[a]Tumor grade (differentiation status) was assessed in 106 samples because the grade for C1 and D11 (severe cirrhosis) could not be determined for Cohort 1. Tumor grade was stratified as I (14 cases) and II–III (75 cases) for the patients in Cohort 2. The clinical information for tumor size and TNM stage was collected only from 82 patients in Cohort 2. The staining for I13 was missing from Cohort 2. The Chi-square test was used for statistical testing. $p < 0.05$ was considered as statistical significance. *$p < 0.05$.

poor prognosis, whereas elevated expression of OTUD5 was associated with extended survival for people with different cancers. Briefly, OTUD5 potentially inhibits tumor progression by regulating PML transcription by deubiquitinating TRIM25 (Fig. 8g).

OTUD5-TRIM25 transcriptionally regulates the expression of the putative tumor suppressor PML, which is dependent on the intact TRIM25 RING domain. OTUD5 inhibits transcriptional activity of TRIM25 by downregulating the ubiquitination level of TRIM25 but the question remains: how exactly does ubiquitination of TRIM25 relate to its functions? It is possible that ubiquitination of TRIM25 serves as a signal, leading to the enrichment of TRIM25 on target genes. Alternatively, ubiquitinated TRIM25 may function as a scaffold to recruit transcriptional repressive cofactors, a function that would be abrogated by OTUD5-mediated deubiquitination. Consistent with these possibilities, OTUD5 cleaves the polyubiquitin chain from TRAF3, an essential type I interferon adapter protein, leading to the disassociation of the adapter protein from a downstream signaling complex and disruption of the type I interferon signaling cascade[18]. Further investigation is needed to understand the precise regulatory mechanism underlying the OTUD5-TRIM25 function.

With deubiquitination activity, OTUD5 functions as a positive regulator of the DNA damage response by antagonizing ubiquitination-induced proteasome-mediated degradation of Ku80 and by stabilizing the UBR5 E3 ubiquitin ligase at damage chromatin[14,15]. OTUD5 also modulates immune responses by deubiquitinating and stabilizing UBR5 to suppress the production of IL-17 and by selectively cleaving the polyubiquitin chains on TRAF3 to regulate IFN-I production[17,18]. The catalytic activity of OTUD5 is necessary for OTUD5 function, which is consistent with our findings that OTUD5 interacts with and deubiquitinates TRIM25 to inhibit tumor growth. Interestingly, OTUD5 activity is dependent on its site-specific phosphorylation at a single residue, Ser177, which is both necessary and sufficient to activate the enzyme[49]. Consistently, our work indicated that phosphorylated OTUD5 is particularly efficient at cleaving the ubiquitin chains of TRIM25. Therefore, modulating OTUD5 phosphorylation may be a promising strategy for tumor treatment.

Our studies reveal a critical role of OTUD5 in tumorigenesis, as indicated through its interaction with TRIM25, suggesting that the OTUD5-TRIM25 axis acts as an important transcriptional regulator in tumor progression. Our studies provide another example of DUB function accompanied by E3 ubiquitin ligases with diverse functional consequences[2,62]. Elegant work carried out by Walsh et al demonstrated that TRIM25 acts as a multifunctional transcriptional factor, lying at the transcription epicenter driving breast cancer metastasis. Defective TRIM25 activity results in broad coordinated changes affecting the expression of many metastatic effectors simultaneously[31,53]. Our work adds a layer to the understanding of TRIM25 regulation, highlighting the importance of TRIM25 ubiquitination in transcriptional regulation.

In summary, our results establish OTUD5 as a pivotal regulator of tumor progression. Our work reveals a layer of regulation for TRIM25-mediated transcriptional suppression. The regulation of TRIM25 by OTUD5 provides a potentially important mechanism that can be targeted for oncotherapy.

## Methods

**Cell culture, chemical treatment, and plasmids.** Hep3B, Huh7, and 293T cells were purchased from the National Infrastructure of Cell Line Resource, and were cultured in Dulbecco's modified Eagle medium supplemented with 10% fetal bovine serum (FBS). All the cell lines have been proven to be negative for mycoplasma contamination. The purchased seed cell lines were freshly thawed and cultured for no >2 months. The cells were kept in a 5% $CO_2$ atmosphere at 37 °C. For trichostatin A (TSA, Sigma) treatment, the cells were plated at 50–70% confluency with 500 nM TSA dissolved in dimethyl sulfoxide. After 24 h, the cells were

**Table 2 OTUD5 was correlated with clinical pathologic characteristics of NSCLC patients.**

| | LUAD OTUD5 | | | LUSC OTUD5 | | |
|---|---|---|---|---|---|---|
| | High | Low | *p*-value | High | Low | *p*-value |
| Age | | | | | | |
| ≥60 | 178 | 185 | 0.530 | 195 | 193 | 0.738 |
| <60 | 72 | 66 | | 42 | 45 | |
| Sex | | | | | | |
| Male | 121 | 119 | 0.860 | 182 | 176 | 0.534 |
| Female | 139 | 141 | | 60 | 66 | |
| Tabaco smoking history | | | | | | |
| Stage3-5 | 154 | 155 | 0.927 | 147 | 180 | 0.001* |
| Stage1-2 | 99 | 98 | | 90 | 57 | |
| Other malignancy history | | | | | | |
| Negative | 209 | 219 | 0.250 | 207 | 223 | 0.027* |
| Positive | 51 | 41 | | 34 | 19 | |
| Laterality | | | | | | |
| Left | 98 | 104 | 0.611 | 94 | 111 | 0.118 |
| Right | 154 | 149 | | 133 | 117 | |
| Location of lung parenchyma | | | | | | |
| Peripheral lung | 65 | 61 | 0.539 | 44 | 47 | 0.687 |
| Central lung | 30 | 34 | | 73 | 70 | |
| Residual tumors | | | | | | |
| Negative (R0) | 173 | 173 | 1.000 | 195 | 191 | 0.307 |
| Positive (R1/R2) | 9 | 9 | | 6 | 10 | |
| Tumor size | | | | | | |
| T1 | 89 | 83 | 0.554 | 65 | 43 | 0.016* |
| T2-T4 | 169 | 176 | | 177 | 199 | |
| Lymph node stage | | | | | | |
| Negative | 177 | 156 | 0.049* | 161 | 146 | 0.152 |
| Positive | 77 | 98 | | 78 | 93 | |
| Distant metastasis | | | | | | |
| Negative | 179 | 173 | 0.205 | 198 | 196 | 0.255 |
| Positive | 9 | 15 | | 2 | 5 | |
| Tumor TNM Stage[a] | | | | | | |
| I–II | 215 | 187 | 0.002* | 160 | 139 | 0.048* |
| III–IV | 41 | 69 | | 80 | 101 | |

The cutoff value for OTUD5 mRNA expression was set at 10.
*NSCLC* non-small-cell lung carcinoma, *LUAD* lung adenocarcinoma, *LUSC* lung squamous cell carcinoma.
[a]Tumor TNM stage was stratified as I-IIA and IIB-IV for the LUSC patients. The Chi-square test was performed to analyze the relationship between OTUD5 expression level and demographic and clinical pathological parameters. All *p*-values were two sided and the level of statistical significance was set at < 0.05. *p < 0.05.

collected for flow cytometric analysis. The full-length complementary DNA (cDNA) and deletion/point mutation mutants of *human OTUD5* were generated by PCR and subcloned into a pCIN4-Flag or pCIN4-HA vector. cDNA or various fragments of the *human TRIM25* gene were cloned into a pRK5-Flag or pCIN4-HA vector for expression.

**RNA interference.** Stable OTUD5- and TRIM25- knockdown cells were generated by infection with lentivirus-based shRNAs. The lentiviral vector (pLVX-shRNA2, Clontech) containing either shRNAs targeting the indicated genes or a negative control of a scrambled sequence was transfected into 293T cells along with psPAX2 and pMD2.G plasmids to produce the lentivirus. The virus containing medium was collected, supplemented with 8 μg ml$^{-1}$ of polybrene (Sigma), and incubated with target cells at 37 °C for 12 h. The infected cells were then subjected to drug selection (3 μg ml$^{-1}$ puromycin). For the siRNA-mediated knockdown assay, the cells were transfected with the appropriate siRNAs using Lipofectamine 3000, and scrambled siRNA was used as a control. After 48 h, the cells were harvested, and the efficiency of the knockdown was verified by immunoblotting. The sequences of shRNA and siRNA used in this study are listed in Supplementary data 2.

**RNA extraction and real-time PCR.** Frozen tissue samples were homogenized using a pestle (Axygen) in a 1.5 ml Eppendorf tube. Total RNA was isolated from tissues or cultured cells using TRIzol reagent (Invitrogen) according to the manufacturer's protocol. Two micrograms of total RNA were reverse transcribed by RevertAid H Minus First Strand cDNA Synthesis kit (Thermo) following the manufacturer's protocol. Quantitative PCR (real-time PCR) was performed in triplicate using SYBR Green mix (Applied Biosystems) and a QuantStudio Dx Real-Time PCR Instrument (Applied Biosystems) under the following conditions: 10 min at 95 °C followed by 40 cycles of 95 °C for 15 s and 60 °C for 1 min. The primers for the RT-PCRs are included in Supplementary data 4.

**Western blot analysis and antibodies.** Western blotting was performed according to a standard protocol, as described previously[63]. The following primary antibodies were used at a dilution of 1:1000: anti-OTUD5 (D8Y2U, Cell Signaling Technology), anti-TRIM25 (Abcam, ab167154), anti-PML (Abcam, ab53773), anti-HA (H9658, Sigma), anti-Flag (F3165, Sigma) and anti-ubiquitin (SC-8017, Santa Cruz). Anti-β-actin (Sigma, AC-15) primary antibody was used at a dilution of 1:5000. Goat anti-Rabbit IgG Secondary Antibody HRP conjugated (SAB Signalway Antibody, L3012) and Goat anti-Mouse IgG Secondary Antibody HRP conjugated (SAB Signalway Antibody, L3032) were used at a dilution of 1:5000.

**Colony formation assay and cell proliferation assay.** A total $1 \times 10^3$ wild-type or target-deleted HCC cells were seeded onto 35 mm dishes in triplicate and incubated for 10 days. The resulting colonies were stained with 2% methylene blue per 50% ethanol for 15 min to be visualized. The results are presented as the averages of data obtained from three independent experiments. The dye in the stained cells were extracted with 1% sodium dodecyl sulfate (SDS) and used for quantifying relative cell growth by spectrophotometer at the absorbance of 600 nm.

**Immunoprecipitation.** The cell lysates were incubated with control IgG or a specific primary antibody overnight at 4 °C. Protein A/G beads were added to lysates and incubated for two hours. The beads were washed three times with lysis buffer. The proteins eluted from the A/G beads were analyzed by western blotting.

**Purification of Flag-tagged proteins.** The Flag-tagged OTUD5 and the substrates for the deubiquitination assays were purified from 293T cells by immunoprecipitation using anti-Flag M2 beads. Forty-eight hours after transfection with expression vectors, the 293T cells were harvested. The cells were lysed in BC500 buffer (20 mM Tris-HCl, pH 7.3; 500 mM NaCl; 20% glycerol; and 0.5% Triton X-100) with brief sonication before overnight incubation with Flag M2 beads at 4 °C. After washing the beads three times with BC100 buffer (20 mM Tris-HCl, pH 7.3; 100 mM NaCl; 20% glycerol; and 0.1% Triton X-100), the purified proteins were eluted by Flag peptide (Sigma).

**Mass spectrometry assay.** The protein complex was purified as described previously with some modifications[48]. The nuclear extracts of the stable Flag-OTUD5/Hep3B cell line were prepared as described above and subjected to a Flag M2 immunoprecipitation. The affinity purified OTUD5-associated proteins were analyzed by liquid chromatography (LC)-MS/MS.

**In vitro deubiquitination assay.** The purified TRIM25-Ubs were incubated with purified OTUD5 (OTUD5/WT, OTUD5/C224S, and OTUD5/S177A) in deubiquitination buffer (50 mM Tris-HCl, pH 8.0; 50 mM NaCl; 1 mM EDTA; 10 mM DTT; and 5% glycerol) for 2 h at 37 °C. The reactions were terminated by adding SDS loading buffer. The mixtures were resolved by sodium dodecyl sulfate polyacrylamide gel electrophoresis for western blot analysis using an anti-ubiquitin antibody.

**RNA-seq.** Hep3B cells were transfected with control siRNA or siRNA targeting TRIM25 for 72 h. Three biological replicates were performed for each group. The cells were then collected in TRIzol reagent, and RNA-seq analysis was performed at Shanghai Majorbio Bio-Pharm Technology Co.,Ltd, China.

**Chromatin immunoprecipitation (ChIP) assay.** Chromatin immunoprecipitation (ChIP) assays were performed according to published protocols[64]. Briefly, cells were fixed with 1% formaldehyde and lysed with ChIP lysis buffer (50 mM Tris-HCl, pH 8.0; 5 mM EDTA; 1% SDS; and 1× protease inhibitor). After sonication and centrifugation, the supernatants were collected in dilution buffer (20 mM Tris-HCl, pH 8.0; 2 mM EDTA; 150 mM NaCl; 1% Triton X-100; and 1× protease inhibitor). Precleaned lysates were incubated with the indicated antibodies overnight at 4 °C before saturated protein A agarose was added to each sample. After extensive washes, the binding components were eluted with 1% SDS and 0.1 M NaHCO$_3$ and reverse cross-linkage was performed at 65 °C for at least 6 h. RT-PCR was performed to detect the relative enrichment of each protein on indicated gene.

**K48-UIM and K63-UIM protein production and identification of ubiquitin chain.** Purified His-tagged K63-UIM and His-tagged K48-UIM were obtained by expressing the proteins in *E. coli*. Bacteria grown in LB medium were induced with 0.25 mM isopropyl-β-d-thiogalactoside (IPTG) for 3 h at 30 °C and then lysed by ultrasound. The K63-UIM and K48-UIM constructs were purified on Ni$^{2+}$-NTA-agarose (Qiagen)[65,66]. Fusion constructs were used for the immunoprecipitation assays with control cells or OTUD5-depleted cells. The eluate was subjected to western blot analysis with anti-TRIM25 antibody.

**Immunofluorescence staining.** For monitoring the formation of PML-NB foci, immunofluorescence staining was performed. The cells were incubated with anti-PML antibody for 2 h at room temperature. The cells were then washed with PBS and incubated with a fluorophore-conjugated secondary antibody. Finally, the cells

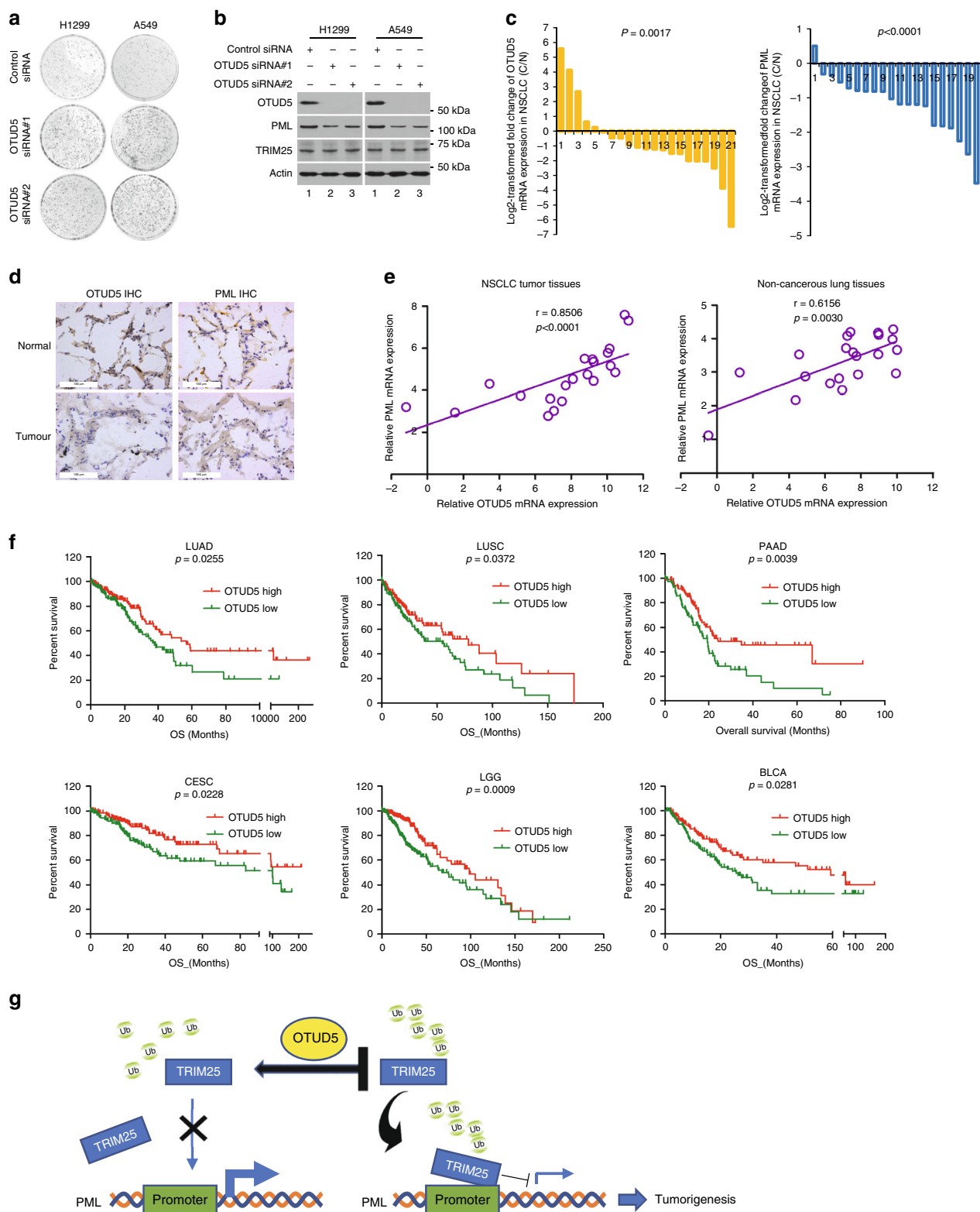

were counterstained with 4′,6-diamidino-2-phenylindole (DAPI) for 5 min to visualize the nuclei with a Zeiss 880 confocal laser-scanning microscope at ×63 magnification.

**Xenograft tumor growth in nude *mice*.** A total of $10^6$ Hep3B-derived cells in a total volume of 100 µl were injected subcutaneously into both flanks of 4- to 6-week-old male nude *mice* (NU/NU; 4–6 weeks old; male; strain 403;Beijing Vital River Laboratory Animal Technology Co., Ltd.). The tumors that formed were monitored every 3 days using a calliper beginning on the sixth day after the

injection. The tumor volume was calculated by the formula $V = \pi \times L \times W^2/6$, where $L$ represents the longest dimension and W the shortest dimension of the tumor[67]. After fifteen days, the *mice* were euthanized, and the tumors were removed and weighed. The nude *mice* were licensed by the Beijing Municipal Committee of Science and Technology. All the *mice* were housed in a temperature-controlled room (22 ± 2 °C) with 40–60% humidity, with a light/dark cycle of 12 h/12 h. All animal experiments were performed according to the animal protocol (No. LA2018013) with Wenhui Zhao approved by the Animal Ethics Committee of the Peking University Health Science Center, China.

**Fig. 8 OTUD5 expression was correlated with clinical features of cancer patients. a** OTUD5 depletion promoted colony formation with lung carcinoma cells. **b** OTUD5 depletion inhibited the expression of PML in H1299 cells and A549 cells. **c** The levels of OTUD5 (left) and PML (right) mRNA were markedly downregulated in the NSCLC samples. Fisher's exact test was used to calculate the $p$-value. $n = 21$. $p = 0.0017$ (OTUD5 mRNA), $p < 0.0001$ (PML mRNA). **d** OTUD5 (left) and PML (right) expression was downregulated in NSCLC tissues (T, magnification 400×) compared with adjacent noncancerous normal tissues (N, magnification 400×) by immunohistochemistry (IHC). IHC staining was repeated twice from two paired tissues. Representative photographs are shown. Scale bar, 100 μm. **e** The expression of OTUD5 was correlated with PML in the NSCLC tissues (left) and paired noncancerous lung tissues (right). Correlation coefficients ($r$) and $p$-values from the Spearman rank-order test are displayed. Each point represents an individual value. $p < 0.0001$ (NSCLC), $p = 0.0030$ (noncancerous tissues). NSCLC, non-small-cell lung carcinoma. **f** OTUD5 expression was correlated with overall survival in *human* cancers. Kaplan–Meier survival curves for overall survival of patients with higher and lower OTUD5 expression using cBioPortal for Cancer Genomics (TCGA) databases. The data for tumor patients from the NSCLC (lung adenocarcinoma, LUAD and lung squamous cell carcinoma, LUSC), bladder urothelial carcinoma (BLCA), pancreatic adenocarcinoma (PAAD), cervical squamous cell carcinoma and endocervical adenocarcinoma (CESC), and brain low-grade glioma (LGG) data sets were extracted and analyzed by log-rank test. $p = 0.0255$ (LUAD), $p = 0.0372$ (LUSC), $p = 0.0039$ (PAAD), $p = 0.0228$ (CESC), $p = 0.0009$ (LGG), $p = 0.0281$ (BLCA). **g** Working model illustrating OTUD5 regulates PML expression and oncogenesis by modulating the ubiquitination and transcriptional activity of TRIM25. Source data are provided as a Source Data file.

**Immunohistochemical staining**. Frozen sections were fixed in 4% formaldehyde for 15 min at room temperature (RT). The sections were washed twice in phosphate buffered saline (PBS) containing 0.3% Triton X-100 for 5 min each time and then blocked with 10% normal goat serum for 1 h at RT. The sections were incubated overnight with OTUD5 (D8Y2U) rabbit mAb (Cell Signaling) at a 1:50 dilution at 4 °C. The sections were further treated with 0.3% $H_2O_2$ for 15 min. After extensive washes, the biotin conjugated secondary antibody was added to the sections. The sections were further incubated for 1 h at RT. After extensive washes, the solution containing streptavidin and biotin conjugated HRP (Beyotime, China) was added to the sections. The sections were incubated for 1 h at RT and then washed in PBS three times for 5 min each time. The slides were developed with 3′-diaminobenzidine (DAB, Beyotime, China) for 10 min at RT followed by rinsing in $H_2O$ for 5 min. The slides were counterstained with hematoxylin, dehydrated, cleared and mounted. The sections were examined by a Leica DM4000 microscope. Protein expression was graded on a scale from "-" to "+++". A grade of "-" indicated no positive cells or the percentage of positive cells was <10%. A grade of "+" indicated a percentage of positive cells of 10%–25%. For an average of positive cells that was ≥26% and ≤49%, the expression was graded as "++", whereas ≥50% positive cells were given a "+++" expression grade. All sections were assessed by two pathologists independently. A third pathologist's assessment was obtained for controversial results.

**Patients**. To determine OTUD5 expression in primary liver cancer, we purchased *human* liver tissue microarray (#BC03119a, ALENA biotechnology, Xi'an, China and #HLivH180Su18, Shanghai Outdo Biotech Company, Shanghai, China). We also assessed OTUD5 expression in 21 non-small-cell lung carcinoma (NSCLC) patients. Patients who underwent curative surgery at The Affiliated Zhangjiagang Hospital of Soochow University from March 2019 to July 2019 were enrolled in this study. Histological confirmation of a primary NSCLC diagnosis was obtained from the Department of Pathology at the hospital. None of the patients had received preoperative adjuvant chemotherapy before surgery. The tumor stage at the time of diagnosis was assessed according to the American Joint Committee on Cancer guidelines (http://www.cancerstaging.org/). Of the NSCLC specimens, matched noncancerous lung tissues samples were obtained from the resected margins far from the originally located tumor site. The tissue samples were immediately stored after surgery at −80 °C for mRNA and immune staining. The demographic and clinical features of all NSCLC patients are shown in Supplementary data 7. Written permission was requested and received from all NSCLC patients in the study. The use of *human* specimens was approved by The Zhangjiagang Hospital Institutional Review Board (No. 2019001).

To study the correlation between OTUD5 level and clinical pathological status and the overall survival of patients with *human* cancers, we used the publicly available clinical information provided (TCGA databases) by the cBioPortal for Cancer Genomics (http://www.cbioportal.org/)[68,69], which included basic demographic and clinical information, as well as survival status after surgery. We retrieved clinical information from non-small-cell lung carcinoma (NSCLC) patients, including 520 patients with lung adenocarcinoma (LUAD) and 484 patients with lung squamous cell carcinoma (LUSC), for the correlation analysis. We extracted tumor cohorts in the higher and lower quartiles for OTUD5 expression to use for the survival analysis of 251 patients with lung adenocarcinoma (LUAD), 240 patients with lung squamous cell carcinoma (LUSC), and 203 patients with bladder urothelial carcinoma (BLCA). We also stratified the data of all patients into higher and lower OTUD5 expression for the survival analysis of 176 patients with pancreatic adenocarcinoma (PAAD), 294 patients with cervical squamous cell carcinoma and endocervical adenocarcinoma (CESC), and 512 patients with brain low-grade glioma (LGG).

**Statistics and reproducibility**. Comparisons of the mean values between groups were analyzed using GraphPad InStat software (Version 5.01, GraphPad Prism, GraphPad Software Inc., San Diego, CA). The statistical significance of the

differences was analyzed using unpaired Student's $t$-test for comparisons of two groups or One-way analysis of variance for comparisons of multiple groups. The data used in this study are presented as the mean ± s.d. Chi-square or Fisher's test was performed to analyze the relationship between OTUD5 expression level and the clinical pathological parameters. Survival curves were plotted using the Kaplan–Meier method and compared by the log-rank test. $p < 0.05$ was considered statistically significant. Detailed information is described in each figure legends. Except for the results from the public database, similar results were obtained from three independent experiments for all other results. Microsoft Excel (Version 11.0) was used to prepare tables and histograms.

**Reporting summary**. Further information on research design is available in the Nature Research Reporting Summary linked to this article.

## Data availability

Statistical source data for the graphical representations and statistical analyses in Figs. 1–8 and Supplementary Figs. 1–5 are provided as a Source Data file. Source data are provided with this paper. The mass spectrometry proteomics data are available via ProteomeXchange with identifier PXD019907. The RNA-seq data is deposited in GEO with the accession code of PRJNA640513. The publicly available mRNA expression and clinical information (TCGA database) were downloaded from the cBioPortal for Cancer Genomics (http://www.cbioportal.org/). Source data are provided with this paper.

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

## Acknowledgements

Wenhui Zhao was supported by the National Natural Science Foundation of China (Grant No. 85141044). Dawei Li is supported by the National Natural Science Foundation of China (Grant No. 81972624) and the Research Foundation for Advanced Talents (Grant No. ZKY201900) by The Affiliated Zhangjiagang Hospital of Soochow University. We are grateful to Dr. Congying Wu for PML-expressing plasmid used in our studies.

## Author contributions

W.Z., D.L., and F.L. were major contributors. Q.S., K.L., L.Z., N.L., K.Y., M.L., N.K., F.T., Z.M., T.L., T.T., J.Q., and W.G. participated in the project. All authors read and approved the final manuscript.

## Competing interests

The authors declare no competing interests.
