## [Peer Review File · Nature Communications]

Reviewers' comments:

Reviewer #1 (Remarks to the Author):

Li et al. present a study that reveals a link between OTUD5 (deubiquitinase) and E3 ubiquitin ligase TRIM25 in the context of oncogenic cellular processes. They show that depletion of OTUD5 results in enhanced proliferation and that this effect is dependent on the interaction with TRIM25. They suggest that OTUD5 specifically deubiquitinates autoubiquitinated TRIM25. Moreover, they link OTUD5 with the expression of tumour suppressor protein PML (TRIM19). Finally, they provide evidence that reduced levels of OTUD5 are associated with an aggressive phenotype and poor clinical outcome of human cancers.

Overall, it is a thorough study with multiple angles and comprehensive results. That said, I have several issues that shed light on the key conclusions of the manuscript.

1. Why there is no signal of OUTD5 and TRIM25 in the inputs in Co-IP experiments presented in Figure 2B and C?

2. How was TRIM25-ubs purified? If the purification was done in the same way described for OUTD5 we cannot be sure that these are truly purified proteins. Even though, high stringency buffer was use there is a possibility that some tight interactions are persistent. This could be dealt with by following a proper protein purification protocol, which authors do not. The SDS page gels of 'purified proteins' should be at least presented.

3. Point 2 brings me to Figure 3. There is no evidence to support that the ubiquitin smear we see on the blots is indeed autoubiquitinated TRIM25. The fact that it most likely is not is reinforced by results presented in Figure 5F where TRIM25 without its RING domain is still heavily ubiquitinated. Thus, the smear could be coming from other proteins, tightly bound by TRIM25.

The only way to discern if the ubiquitin smear is detecting autoubiquitinated TRIM25 is to perform similar experiments with the TRIM25 K117 mutant. K117 is the key residue responsible for autoubiquitination.

This also brings me to the discussion section (lanes 402 - 404. 'It is possible that ubiquitination on the RING domain of TRIM25 makes structural transform or behaves as a signal to help the enrichment of TRIM25 on the target genes.' In the light of what we know about TRIM25 autoubiquitination this speculation (without proper control) is ungrounded.

If the authors can show that the TRIM25 K117R is phenocopying OUTD5 this will be a substantial discovery as thus far the role TRIM25 autoubiquitination remains unknown.

4. The TRIM25 K117R mutant should be used in other assays, including CHIP presented in Figure 5G. Currently, the only conclusion that we can draw from this experiment is that catalytically dead TRIM25 is not enriched at the PML TSS. The only way to prove that this is due to the OUTD5-dependent deubiquitination of TRIM25 would be with the above mentioned mutant.

5. The RNAseq as well as Mass Spectrometry should be publicly available. This is very important as the authors base the whole manuscript on a relatively weak mass spectrometry result (2 unique peptides of TRIM25 in the OUTD5 IP).

Minor comments

1. Wine 69 should read '...innate immunity'

2. Are the transcripts upregulated upon TRIM25 knockdown detected by RT-PCR (Fig.4G) also

upregulated in RNAseq experiment (Fig. 4C). If so they should be indicated in the same way as PML.

Reviewer #2 (Remarks to the Author):

In the manuscript entitled "OTUD5 cooperates with TRIM25 in transcriptional regulation and tumor progression via the deubiquitination activity" by F. Li et al., the authors investigated the role of the OTUD deubiquitinases in cancer, in particular growth suppressive effects of OTUD5 on hepatocarcinoma cell lines.

Mechanistically, the authors propose that OTUD5 controls the deubiquitination of TRIM25, inhibiting its repressive effects on PML expression. The latter would account for the increase in tumor cell growth induced by OTUD5 knock down (KD).

TCGA database analyses shows that low expression of OTUD5 is associated with a decrease in overall survival of patients with six different types of carcinoma (not hepatocarcinomas). Although, in principle, this is an important observation, the role of OTUD5 in tumor development should have been assessed in more physio-pathological relevant models than merely cell lines.

The authors clearly established that OTUD5 interacts and deubiquitinylates TRIM25, and that OTUD5 KD increases tumor size in hepatocarcinoma cell line xenografts. However, the role of TRIM25 deubiquitination in the latter is only correlative. While anti-tumoral properties of PML are well known in many cancers, its involvement in tumor growth as a target of OTUD5-mediated TRIM25 deubiquitination was not clearly established in this study.

Main points

My main concern is linked to the cellular models. The authors based their study on 2 hepatocarcinoma cell lines, mainly used in culture. Cross talks between OTUD5 and TRIM25/PML should have been investigated (by RNA seq, ChIP, TRIM25 ubiquitination etc.) directly from mice, at least from the xenografted cells, more relevant for tumorigenesis. Double KD for OTUD5 and TRIM25 or PML in xenografts are also required to validate the mechanism of action proposed by the authors for OTUD5 function in tumor progression.

Since OTUD5 was shown to target various substrates, RNAseq analysis of OTUD5 KD cells should be performed and compared to TRIM25 KD to determine overlapping and non-overlapping pathways/ genes more broadly.

GSEA/GO analyses would provide information on the pathways targeted by TRIM25KD in these cell lines compared to other models, as those found by TRIM25 ChIPseq and RNAseq upon TRIM25 KD in breast cancer in vivo models.

The pattern of PML protein migrations presented here, in these cell lines, is quite unusual. PML is expressed as various isoforms and the most abundant one is PML-I with a size >120 kDa. Here, all WB analyses shows a band close to 75kDa. How do the authors explain this difference? Do other anti-PML antibodies show the same pattern, especially antibodies specific for different PML isoforms?

In figure 4D, the authors focused on the expression of genes potentially regulated by PML (such as CDKN1A or BAX), showing by qRT-PCR significant induction upon TRIM25 KD. Are these transcripts also found upregulated in the RNAseq analysis?

To really assess any PML involvement in this TRIM25 targets, the authors should KD PML together with that of TRIM25 and assess expression of those genes, or more broadly on TRIM25 targets.

It is not clear why the authors used TSA to activate apoptosis. P53 is mutated in these two cell lines. A more classical way would have been to use TNF or irradiation known to activate a PML-

dependent apoptosis. More importantly, a KD of a pro-apoptotic gene is expected to increase cell viability, this does not mean that it is specifically involved in TRIM25 KD effects. The authors would have to test the effect of PML KD alone on cell survival and compare to the double KD. Propidium Iodide staining is indicative for cell survival rather than apoptosis.

Although PML increase upon TRIM25 siRNA in cell lines is clear, the mechanism proposed by the author is not. Using ChIP-PCR (upon HA-TRIM25 or HA-IRF8 overexpression), the authors proposed that TRIM25 binds PML TSS and competes with IRF8-binding on interferon responsive elements within PML promoter. However, the effect of TRIM25 shRNA on HA-IRF8 enrichment at PML promoter was far too low to explain PML upregulation. In addition, although OTUD5 KD increased TRIM25 binding onto PML promoter, it only slightly decreased in PML mRNA level (Fig 6C and D).

Since PML is heavily regulated by SUMO and ubiquitin conjugation, the authors should investigate whether the TRIM25 ubiquitin-ligase and the OTUD5 DUB could affect PML stability by directly regulating its ubiquitylation.

More globally, this is not clear whether OTUD5 only affects TRIM25 ubiquitylation or its ubiquitin ligase activity and which of the two is involved in TRIM25 cell growth effects. Indeed, the authors only assessed trim25 auto-ubiquitination as reflecting its ubiquitin ligase activity, while other TRIM25 protein targets may be also deubiquitinated within OTUD5 complex. The authors could test at least global ubiquitination in cells expressing HA-TRIM25 +/- OTUD5.

TRIM25 mutants for its ubiquitylated lysines should be compared to delta-RING mutant, both in colony formation or PML regulation. The authors could check which type of ubiquitin chains is affected by the DUB, in particular in vivo, using K63 or K48 chain specific antibodies in their IP/WB analyses.

Similarly, the role of OTUD5 DUB activity was investigated neither on PML nor on tumor growth in xenografts using OTUD5 mutant for its catalytic cysteine or UIM. Thus, the title overstates the role of DUB activity in tumor progression.

Finally, it is not clear whether the authors propose a general mechanism of OTUD5/TRIM25 function in cancer or a specific one to hepatocarcinoma. Assessing OTUD5 in the metastatic function of TRIM25, in breast cancer models, is particularly important to conclude on any role of OTUD5 in "tumor progression", as stated in the title.

Minor points

While PML mRNA expression was 3 fold-induced by acute down-regulation of TRIM25 using siRNA (Fig 4c and D), TRIM25 KD had very low effects shRNA in fig 5H, where shRNA were apparently used. The authors did mention the shRNA before, why do the authors switch to sh for some experiments?

Quantifications on the total PML protein amount is required to validate the effects of the double TRIM25/OTUD5 KD onto PML expression and conclude that OTUD5-induced down regulation of PML is TRIM25 dependent.

Since PML is an IFN-target gene and OTUD5 was shown to down regulate IFN type I response, up regulation of IFN pathway in OTUD5 KD might have opposite effect on PML expression that would even limit PML down regulation upon OTUD5 KD. How the authors take that into account? Is there any IFN pathways signature activated by OTUD5 KD in these cell lines?

Reviewer #3 (Remarks to the Author):

OTUD5 cooperates with TRIM25 in transcriptional regulation and tumor progression via the deubiquitination activity

This interesting new manuscript describes a ubiquitin based signaling axis that controls proliferation, gene expression and which likely plays a role in cancer. It centers on a deubiquitinase OTUD5, which the authors suggest regulates the ubiquitination of a transcriptional repressor TRIM25. This in turn control the expression of PML, among other genes, to control tumorigenesis. The manuscript was well organized, the data was clear, logical and well put together, and the findings were new and interesting. I am therefore enthusiastic about the eventual impact of the manuscript and supportive of its publication. Below are a few points and suggested experiments that I would hope the reviewers can first address prior to full acceptance of the paper.

Major points

1. In figure 3, the authors make the case that TRIM25 ubiquitination is controlled by OTUD5. However, in all of the panels in this experiment, the authors are not directly assessing the covalent modification of TRIM25 with ubiquitin, and instead, rely on TRIM25 pulldowns followed by ubiquitin immunoblots. Thus, it is unclear if the smeared band in the pulldown is in fact TRIM25, or some other co-precipitating protein. I understand that these are routinely done under near-denaturing conditions, but this cannot rule out this concern. Since this is a top-line conclusion of the manuscript, it must be addressed. I do not think all of the experiment needs to be redone, but just one or two, to show convincingly that it is in fact TRIM25 that is ubiquitin modified. I can imagine this being done in two ways. First, they could blot the TRIM25 pulldowns for TRIM25 (or its tag) to show that it in fact smears into high-molecular weight bands. Alternatively, the authors can pulldown ubiquitin and show that TRIM25 comes with it.
2. The authors suggest that the data in Fig 2 shows that the increased growth of OTUD5 depleted cells is TRIM25 dependent. However, these data are inconclusive in this respect. The depletion of TRIM25 reduces proliferation in cells with OTUD5, and the relative decrease in proliferation between control and OTUD5-depleted cells that are depleted for TRIM25 appears almost the same. I have no issue with the result, but believe it is being over-interpreted in this case.
3. The authors refer to OTUD5 as a tumor suppressor in several places. The bar for calling it that is, I think, quite high. Since there is no good evidence that TRIM25 is lost or mutated in tumors, and by itself, would cause tumors, I think it would be more accurate to say that it has tumor suppressive properties. Not everything that suppresses proliferation is a tumor suppressor.
4. Since the ubiquitination of trim25 appears to not be degradative it would be interesting to know what the chain linkage is, or to at least show that it is not K48 linked. This could be done using a K48R mutated version of ubiquitin.
5. The mechanism for how trim25 is regulated is unclear. Although not necessary to figure out for publication, I would suggest simply testing if it might affect TRIM25 nuclear localization or chromatin binding. These experiments could be done easily, using existing reagents, with simple fractionation techniques.
6. The authors say that they tested to see if trim25 regulates PML and that it doesn't, but the data is not shown. This is actually a rather important finding and the negative data should be included.
7. In a couple of places, the authors performed chip experiments and used IgG IPs in untransfected cells as a control for an HA IP in a transfected cell population. These should actually be done using an HA IP in control transfected cells. See figure 5 in particular.
8. Figure 3B is not well controlled since phosphatase inhibitors are only used to control some of these conditions. Because these will block all dephosphorylation it makes the interpretation very difficult.

Minor points

1. Would be interested to test OTUD5 overexpression in some of these assays.

2. The inputs in Fig 2A don't make any sense. There shouldn't be three inputs for this experiment. The authors need to clarify and explain better how this is done.
3. Fig 2G- need to show TRIM25 in the input of the IP to interpret the result that the UIM mediates binding partially.
4. The authors should be more clear that the phosphorylation site has already been shown to modulate OTUD5 activity.
5. Authors should indicate that OTUD5 is also called DUBA.
6. Need show evidence that the TRIM25 shRNA works
7. Typo in fig 5H (mmRNA levels)
8. In figure 5 there is some confusion of the use of flag or ha tags between the text and figure
9. Remove 2F and 5E- they are not needed

Reviewers' comments:

Reviewer #1 (Remarks to the Author):

Li et al. present a study that reveals a link between OTUD5 (deubiquitinase) and E3 ubiquitin ligase TRIM25 in the context of oncogenic cellular processes. They show that depletion of OTUD5 results in enhanced proliferation and that this effect is dependent on the interaction with TRIM25. They suggest that OTUD5 specifically deubiquitinates autoubiquitinated TRIM25. Moreover, they link OTUD5 with the expression of tumour suppressor protein PML (TRIM19). Finally, they provide evidence that reduced levels of OTUD5 are associated with an aggressive phenotype and poor clinical outcome of human cancers.

Overall, it is a thorough study with multiple angles and comprehensive results. That said, I have several issues that shed light on the key conclusions of the manuscript.

Response: The authors greatly appreciate the referee for his positive, constructive and detailed comments and suggestions on our manuscript. We have tried our best to revise our manuscript according to these comments. We hope that this revised manuscript is now suitable for publication.

1. Why there is no signal of OUTD5 and TRIM25 in the inputs in Co-IP experiments presented in Figure 2B and C?

1. Response: The authors greatly appreciate the referee's critical suggestion to improve the quality of the manuscript. We apologize for the confusion. In the old manuscript, the labels of figure 2B and 2C were not suitable, because we only examined the elute of IP assay and tried to confirm that OTUD5 and TRIM25 had been pulled down by the antibodies. We are very thankful for this criticism. We have re-done the western blotting assay and presented in the revised figures (Fig. 2B and 2C).

2. How was TRIM25-ubs purified? If the purification was done in the same way described for OUTD5 we cannot be sure that these are truly purified proteins. Even though, high stringency buffer was use there is a possibility that some tight interactions

are persistent. This could be dealt with by following a proper protein purification protocol, which authors do not. The SDS page gels of 'purified proteins' should be at least presented.

2. Response: The authors greatly appreciate the referee's comment on our manuscript. TRIM25-ubs was purified according to the Method named "Purification of Flag-tagged proteins". We have amended the description to avoid misunderstanding (page 20: "substrates" to "the substrates for deubiquitination assays"). Moreover, we confirmed that the ubiquitin smear was indeed ubiquitinated TRIM25 by assaying the elute with antibodies against Flag (Fig. 3F) and HA (Fig. 5E).

3.1 Point 2 brings me to Figure 3. There is no evidence to support that the ubiquitin smear we see on the blots is indeed autoubiquitinated TRIM25. The fact that it most likely is not is reinforced by results presented in Figure 5F where TRIM25 without its RING domain is still heavily ubiquitinated. Thus, the smear could be coming from other proteins, tightly bound by TRIM25.

3.1 Response: The authors greatly appreciate the reviewer for critical and constructive comments and suggestions to improve the quality of the manuscript. We exchanged the image with a short-exposure WB image (Fig. 5E). Furthermore, we analyzed the elute by western blotting using anti-HA antibody. Consistently, RING domain-deletion significantly reduced, but not eliminated the ubiquitin chain of TRIM25 (Fig. 5E).

3.2 The only way to discern if the ubiquitin smear is detecting autoubiquitinated TRIM25 is to perform similar experiments with the TRIM25 K117 mutant. K117 is the key residue responsible for autoubiquitination.

3.2 Response: The authors greatly appreciate the reviewer for the constructive suggestions to improve the quality of the manuscript. We examined the ubiquitination level of TRIM25 K117R mutant. Indeed, K117R mutant almost eliminated the autoubiquitination of TRIM25 (Supplementary fig. 2A).

3.3 This also brings me to the discussion section (lanes 402 - 404. 'It is possible that ubiquitination on the RING domain of TRIM25 makes structural transform or behaves as a signal to help the enrichment of TRIM25 on the target genes.' In the light of what we know about TRIM25 autoubiquitination this speculation (without proper control) is ungrounded.

3.3 Response: The authors greatly appreciate the referee's comment for improving the quality of our manuscript. We tuned-down the conclusion in the revised manuscript. We changed the expression "It is possible that ubiquitination on the RING domain of TRIM25 makes structural transform or behaves as a signal to help the enrichment of TRIM25 on the target genes." to "It is possible that ubiquitination of TRIM25 behaves as a signal to help the enrichment of TRIM25 on the target genes." (page 17).

3.4 If the authors can show that the TRIM25 K117R is phenocopying OUTD5 this will be a substantial discovery as thus far the role TRIM25 autoubiquitination remains unknown.

3.4 Response: The authors greatly appreciate the referee's suggestion to improve the quality of the manuscript. In the overexpression assay, we demonstrated that co-overexpression with OTUD5 significantly reduced the ubiquitin chain of TRIM25, which was consistent with the result of TRIM25-K117R mutant overexpression (Supplementary fig. 2A).

4. The TRIM25 K117R mutant should be used in other assays, including CHIP presented in Figure 5G. Currently, the only conclusion that we can draw from this experiment is that catalytically dead TRIM25 is not enriched at the PML TSS. The only way to prove that this is due to the OUTD5-dependent deubiquitination of TRIM25 would be with the above mentioned mutant.

4. Response: The authors greatly appreciate the referee's critical suggestion to improve the quality of the manuscript. We performed ChIP assay with plasmid encoding TRIM25 K117R. The result demonstrates that K117R mutant of TRIM25 reduced the accumulation of TRIM25 at the PML TSS (Supplementary fig. 2B).

5. The RNAseq as well as Mass Spectrometry should be publicly available. This is very important as the authors base the whole manuscript on a relatively weak mass spectrometry result (2 unique peptides of TRIM25 in the OUTD5 IP).

5. Response: The authors greatly appreciate the referee's critical suggestion to improve the quality of the manuscript. We have attached the results of the mass spectrometry (Supplementary table 1) and the RNAseq (Supplementary table 3).

Minor comments

1. Wine 69 should read '...innate immunity'

1.Response: The authors greatly appreciate the referee's critical comments to improve the quality of the manuscript. We have amended our description according to the suggestion (page 3).

2. Are the transcripts upregulated upon TRIM25 knockdown detected by RT-PCR (Fig.4G) also upregulated in RNAseq experiment (Fig. 4C). If so they should be indicated in the same way as PML.

2.Response: The authors greatly appreciate the referee's critical comments to improve the quality of the manuscript. We checked the results of RNA-seq experiment. The transcripts of BAX were significantly upregulated upon TRIM25 depletion, and we have indicated it in Figure 4C. The change of the other transcripts was not significant.

Reviewer #2 (Remarks to the Author):

In the manuscript entitled “OTUD5 cooperates with TRIM25 in transcriptional regulation and tumor progression via the deubiquitination activity” by F. Li et al., the authors investigated the role of the OTUD deubiquitinases in cancer, in particular growth suppressive effects of OTUD5 on hepatocarcinoma cell lines. Mechanistically, the authors propose that OTUD5 controls the deubiquitination of TRIM25, inhibiting its repressive effects on PML expression. The latter would account for the increase in tumor cell growth induced by OTUD5 knock down (KD). TCGA database analyses shows that low expression of OTUD5 is associated with a decrease in overall survival of patients with six different types of carcinoma (not hepatocarcinomas). Although, in principle, this is an important observation, the role of OTUD5 in tumor development should have been assessed in more physio-pathological relevant models than merely cell lines.

The authors clearly established that OTUD5 interacts and debubiquitylates TRIM25, and that OTUD5 KD increases tumor size in hepatocarcinoma cell line xenografts. However, the role of TRIM25 deubiquitination in the latter is only correlative. While anti-tumoral properties of PML are well known in many cancers, its involvement in tumor growth as a target of OTUD5-mediated TRIM25 deubiquitination was not clearly established in this study.

Response: Thank you for your response and the referees' positive and constructive comments and suggestions on our manuscript. We have extensively and substantively revised the manuscript in response to the insightful comments. We have worked diligently to obtain new evidence demonstrating that OTUD5-TRIM25 axis plays an important role in tumorigenesis.

Importantly, we re-designed the xenograft assay accordingly. The results demonstrated that OTUD5 depletion indeed promoted tumor growth. Meanwhile, TRIM25 knockdown inhibited growth-promoting effort of OTUD5 knockdown. In addition, OTUD5/TRIM25 double knockdown cells formed smaller tumor sphere

than that formed by OTUD5-depleted cells, demonstrating that OTUD5-TRIM25 axis plays an important role in the regulation of tumor growth (Fig. 7A-B).

Furthermore, we think the mechanism we supported is a general mechanism in tumor progression. We performed colony formation assay with lung cancer cell lines (H1299 and A549), wherein OTUD5 knockdown promoted cell proliferation in both H1299 cells and A549 cells. Additionally, we collected more Non-small-cell lung carcinoma (NSCLC) tissues with adjacent paired non-cancerous tissues. The mRNA level of OTUD5 was markedly decreased in tissues from Non-small-cell lung carcinoma (NSCLC) compared with adjacent paired non-cancerous tissues (Fig. 8C left). Consistently, the immunostaining of OTUD5 was observed in normal lung and tumor cells, with higher staining occurred in the control cells than in the cancerous cells (Fig. 8D left). To assess OTUD5 mediated PML regulation in tumor development in more physio-pathological relevant conditions, we also investigated PML expression and its correlation with OTUD5 level in NSCLC. PML expression was markedly downregulated in tumor tissues in comparison with adjacent non-cancerous tissues (Fig. 8C right and D right). PML staining was detected in both cytoplasm and nucleus of epithelial cells from tumor and non-tumorous tissues (Fig. 8D right). Importantly, OTUD5 expression was significantly positively correlated with PML level in both tumor and non-tumorous tissues (Fig. 8E), supporting that OTUD5 may act as general regulator of tumor progression.

We found your comments very useful in improving the quality of our manuscript in a more convincing and precise way. In this revised manuscript, we have incorporated a considerable amount of additional experimental evidence to fully address the concerns. We hope that this revised manuscript is now suitable for publication.

Main points

1. My main concern is linked to the cellular models. The authors based their study on 2 hepatocarcinoma cell lines, mainly used in culture. Cross talks between OTUD5 and TRIM25/PML should have been investigated (by RNA seq, ChiP, TRIM25 ubiquitination

etc.) directly from mice, at least from the xenografted cells, more relevant for tumorigenesis. Double KD for OTUD5 and TRIM25 or PML in xenografts are also required to validate the mechanism of action proposed by the authors for OTUD5 function in tumor progression.

1. Response: The authors greatly appreciate the referee for critical suggestion to improve the quality of the manuscript. We re-designed the xenograft assay with OTUD5 and TRIM25 or PML double knockdown cells. TRIM25 knockdown inhibited the growth-promoting effort of OTUD5 knockdown, and TRIM25 depletion could no longer inhibit tumor growth when PML was depleted. Moreover, we investigated the mRNA level of PML and BAX in the xenograft cells. The results revealed that OTUD5 knockdown reduced expression of PML gene and BAX gene (Fig. 7A-C).

2. Since OTUD5 was shown to target various substrates, RNAseq analysis of OTUD5 KD cells should be performed and compared to TRIM25 KD to determine overlapping and non-overlapping pathways/ genes more broadly.

2. Response: The authors greatly appreciate the referee for this suggestion to improve the quality of the manuscript. To evaluate the overlapping functions of OTUD5 and TRIM25, we performed RNA sequencing analysis with OTUD5-depleted cells. Gene ontology (GO) analysis on the overlapping genes demonstrated that OTUD-TRIM25 axis play a role in many critical cellular signaling cascades, such as cell cycle and cell division processes, implying OTUD5-TRIM25 axis may affect cell proliferation by regulating cell cycle (Fig. 4M-N). Additionally, PML is one of the downregulated genes upon OTUD5 depletion (Supplementary table. 3).

3. GSEA/GO analyses would provide information on the pathways targeted by TRIM25KD in these cell lines compared to other models, as those found by TRIM25 ChIPseq and RNAseq upon TRIM25 KD in breast cancer in vivo models.

3. Response: The authors greatly appreciate the referee's critical suggestion to improve the quality of the manuscript. GO analyses upon our results

demonstrated that TRIM25 knockdown play a role in many critical cellular signaling cascades, such as cell cycle, regulation of apoptosis and regulation of cell proliferation (Supplementary fig. 1B). Previous report reveals that the enriched biological processes of TRIM25 targets were dominated by developmental processes, suggesting that TRIM25 may mediate metastases by affecting differentiation in breast cancer model (Walsh et al. 2017, Cell reports).

4.The pattern of PML protein migrations presented here, in these cell lines, is quite unusual. PML is expressed as various isoforms and the most abundant one is PML-I with a size >120 kDa. Here, all WB analyses shows a band close to 75kDa. How do the authors explain this difference? Do other anti-PML antibodies show the same pattern, especially antibodies specific for different PML isoforms?

4. Response: The authors greatly appreciate the referee's critical suggestion to improve the quality of the manuscript. We apologize for the confusion. The anti-PML antibody used in our research was purchased from Abcam (ab53773) which was used for detecting PML with a size >100 kDa. Because the signal intensity of 75 kDa label of the protein marker that we used in our western blotting assay is very strong, we preferred to label 75 kDa labels. Indeed, the band size of PML was larger than 100 kDa, and we have amended all of the labels in the figures according to the suggestion.

5.In figure 4D, the authors focused on the expression of genes potentially regulated by PML (such as CDKN1A or BAX), showing by qRT-PCR significant induction upon TRIM25 KD. Are these transcripts also found upregulated in the RNAseq analysis?

5.Response: The authors greatly appreciate the referee's suggestion. We checked the results of RNA seq. Consistently, the transcript of BAX was significantly upregulated upon TRIM25 depletion, and we have indicated it in figure 4C. The upregulation of CDKN1A in the RNA-seq analysis was not significant.

6.To really assess any PML involvement in this TRIM25 targets, the authors should KD PML together with that of TRIM25 and assess expression of those genes, or more broadly on TRIM25 tagets.

6. Response: The authors greatly appreciate the referee's comment for improving the quality of our manuscript. We performed RT-PCR assay with TRIM25/PML double knocked down cells. TRIM25 knockdown no longer upregulated repression of BAX in PML-depleted cells (Fig. 4I).

7. It is not clear why the authors used TSA to activate apoptosis. P53 is mutated in these two cell lines. A more classical way would have been to use TNF or irradiation known to activate a PML-dependent apoptosis. More importantly, a KD of a pro-apoptotic gene is expected to increase cell viability, this does not mean that it is specifically involved in TRIM25 KD effects. The authors would have to test the effect of PML KD alone on cell survival and compare to the double KD. Propidium Iodide staining is indicative for cell survival rather than apoptosis.

7. Response: The authors greatly appreciate the referee's constructive comments to improve the quality of our manuscript. We performed the TSA-induced apoptosis assay according to the research reported by Steven G. Gray et al (Steven G. Gray et al, 1999, Experimental cell research). We thought that it was hard to draw a conclusion from one assay. The apoptosis assay in our manuscript demonstrated that TRIM25 knockdown promoted cell arrested at sub-G1, and that TRIM25 knockdown no longer increased the number of sub-G1 cells when PML was depleted. We tuned-down the conclusion to make it more precise. We are very thankful for your criticisms to improve our manuscript.

8. Although PML increase upon TRIM25 siRNA in cell lines is clear, the mechanism proposed by the author is not. Using ChIP-PCR (upon HA-TRIM25 or HA-IRF8 overexpression), the authors proposed that TRIM25 binds PML TSS and competes with IRF8-binding on interferon responsive elements within PML promoter. However, the effect of TRIM25 shRNA on HA-IRF8 enrichment at PML promoter was far too low to explain PML upregulation. In addition, although OTUD5 KD increased TRIM25 binding onto PML promoter, it only slightly decreased in PML mRNA level (Fig 6C and D).

8. Response: The authors greatly appreciate the referee's positive, constructive and detailed comments and suggestions on our manuscript. Firstly, the

mechanism might be very important, although the rangeability was low. Secondly, there were possibly some other transcriptional factors of PML expression. We agree with the referee's idea that it is exaggerated to propose that TRIM25 repressed PML expression through inhibiting the recruitment of IRF-8 to the promoter of PML. Hence, we tune-down the conclusion in the revised manuscript. We change the expression to "These data suggested that TRIM25 repressed PML expression possibly through inhibiting the recruitment of IRF-8 to the promoter of PML, at least partly." (page 12).

9. Since PML is heavily regulated by SUMO and ubiquitin conjugation, the authors should investigate whether the TRIM25 ubiquitin-ligase and the OTUD5 DUB could affect PML stability by directly regulating its ubiquitylation.

9. Response: The authors greatly appreciate the referee's critical and constructive comments and suggestions to improve the quality of the manuscript. We performed PML ubiquitination assay, and found that co-overexpression with TRIM25 moderately increased ubiquitination level of PML (Supplementary fig. 4A). Moreover, we performed *in vitro* deubiquitination assay with OTUD5 using PML-ubs as substrate. The results showed that OTUD5 could not cut ubiquitin chain of PML *in vitro* (Supplementary fig. 4B).

10. More globally, this is not clear whether OTUD5 only affects TRIM25 ubiquitylation or its ubiquitin ligase activity and which of the two is involved in TRIM25 cell growth effects. Indeed, the authors only assessed trim25 auto-ubiquitination as reflecting its ubiquitin ligase activity, while other TRIM25 protein targets may be also deubiquitinated within OTUD5 complex. The authors could test at least global ubiquitination in cells expressing HA-TRIM25 +/- OTUD5.

10. Response: The authors greatly appreciate the referee's comments to improve our manuscript. We performed *in vivo* ubiquitination assays with two of the TRIM25 targets, SFN (Urano, T. et al, 2002, Nature) and MAVS (Castanier C. et al, 2012, BMC Biol). We found that co-overexpression with OTUD5 would reduce the ubiquitination levels of SFN and MAVS, implying OTUD5-TRIM25 axis may have

broader functions in some other cellular processes (Supplementary fig. 5).

11. TRIM25 mutants for its ubiquitinated lysines should be compared to delta-RING mutant, both in colony formation or PML regulation. The authors could check which type of ubiquitin chains is affected by the DUB, in particular in vivo, using K63 or K48 chain specific antibodies in their IP/ WB analyses.

11. Response: The authors greatly appreciate the referee's the constructive suggestions to improve the quality of the manuscript. We performed ChIP assay with plasmid encoding TRIM25/K117R mutant. Consistently, accumulation of TRIM25/K117R mutant was significantly less than TRIM25-WT (Supplementary fig. 2B). We performed K48-UIM/K63-UIM IP assay with Hep3B cells, described in the method section. The results showed that OTUD5 depletion increased K63 ubiquitin chain of TRIM25 (Supplementary fig. 3).

12. Similarly, the role of OTUD5 DUB activity was investigated neither on PML nor on tumor growth in xenografts using OTUD5 mutant for its catalytic cysteine or UIM. Thus, the title overstates the role of DUB activity in tumor progression.

Response: The authors greatly appreciate the referee's the constructive suggestions to improve the quality of the manuscript. We performed colony formation assay with OTUD5/C224S, and found that overexpression with OTUD5/WT but not OTUD5/C224S significantly inhibit cell proliferation (Fig. 3H).

13. Finally, it is not clear whether the authors propose a general mechanism of OTUD5/TRIM25 function in cancer or a specific one to hepatocarcinoma. Assessing OTUD5 in the metastatic function of TRIM25, in breast cancer models, is particularly important to conclude on any role of OTUD5 in "tumor progression", as stated in the title.

Response: The authors greatly appreciate the referee's the constructive comments and suggestions. We think the mechanism we supported is a general mechanism in tumor progression. The function of TRIM25 in breast cancer progression was studied broadly and well established. Previous report reveals that the enriched biological processes of TRIM25 targets were dominated by

developmental processes, suggesting that TRIM25 may mediate metastases by affecting differentiation in breast cancer model (Walsh et al. 2017, Cell reports). Gene ontology (GO) analysis on OTUD5-TRIM25-overlapping genes demonstrated that OTUD5-TRIM25 axis play a role in many critical cellular signaling cascades, such as cell cycle and cell division processes, implying OTUD5-TRIM25 axis may affect cell proliferation by regulating cell cycle (Fig. 4M-N). We performed colony formation assays with lung cancer cells line, H1299 and A549. The results demonstrated that OTUD5 knockdown promoted cell proliferation (Fig. 8A). Moreover, the mRNA level of OTUD5 was markedly decreased in tissues from Non-small-cell lung carcinoma (NSCLC) compared with adjacent paired non-cancerous tissues (Fig. 8C left). Consistently, the immunostaining of OTUD5 was observed in normal lung and tumor cells, with higher staining occurred in the control cells than in the cancerous cells (Fig. 8D left). We then investigated PML expression and its correlation with OTUD5 level in NSCLC. PML expression was markedly downregulated in tumor tissues in comparison with adjacent non-cancerous tissues (Fig. 8C right and D right). Importantly, OTUD5 expression was significantly positively correlated with PML level in both tumor and non-tumorous tissues (Fig. 8E), supporting that OTUD5 may regulate PML in NSCLC patients. Further, we assessed the correlation between OTUD5 expression and clinical pathological status and patients' survival using TCGA database. The OTUD5 expression was significantly correlated with tumor size, lymph node invasion and TNM stage in NSCLC patients (Table 2). Additionally, the patients with elevated OTUD5 expression had significantly longer overall survival compared to the patients bearing tumors that express lower levels of OTUD5 transcript in NSCLC (LUAD and LUSC), PADD, CESC, LGG and BLCA (Fig. 8F). Taken together, these findings indicated that OTUD5 displayed strong tumor suppressive properties with functional consequences in different tumor suppression.

Minor points

1. While PML mRNA expression was 3 fold-induced by acute down-regulation of TRIM25 using siRNA (Fig 4c and D), TRIM25 KD had very low effects shRNA in fig 5H,

where shRNA were apparently used. The authors did mention the shRNA before, why do the authors switch to sh for some experiments?

1.Response: The authors greatly appreciate the referee's constructive comments. As you can see, we transfected the cell with expressing plasmids in these assays. It is easier to handle a single-transfection assay than to handle a plasmid/siRNA co-transfection assay.

2.Quantifications on the total PML protein amount is required to validate the effects of the double TRIM25/OTUD5 KD onto PML expression and conclude that OTUD5-induced down regulation of PML is TRIM25 dependent.

2.Response: The authors greatly appreciate the referee's constructive suggestions to improve the quality of the manuscript. We performed quantifications on the PML protein of each bound using imagelab software (Fig. 6E). The results support our conclusion.

3.Since PML is an IFN-target gene and OTUD5 was shown to down regulate IFN type I response, up regulation of IFN pathway in OTUD5 KD might have opposite effect on PML expression that would even limit PML down regulation upon OTUD5 KD. How the authors take that into account? Is there any IFN pathways signature activated by OTUD5 KD in these cell lines?

3.Response: The authors greatly appreciate the referee's the suggestions on our manuscript. We believe OTUD5 has broad functions in many important cellular processes. For example, our previous research (Li. et al. 2019, CMLS) and the study did by Angelo de Vivo et al. (2019, NAR) showed that OTUD5 play an important role in DNA repair response. The researches about OTUD5 targeting IFN pathway (Nobuhiko Kayagaki et al.2007, Science; Sascha Rutz et al. 2015, Nature.) and that PML is an IFN-target gene (Stadler, M et al, 1995, Oncogene.) are very great. It is hard for us to figure out the overall mechanism in one signal manuscript. We will continue to work on the function of OTUD5.

Reviewer #3 (Remarks to the Author):

OTUD5 cooperates with TRIM25 in transcriptional regulation and tumor progression via the deubiquitination activity

This interesting new manuscript describes a ubiquitin based signaling axis that controls proliferation, gene expression and which likely plays a role in cancer. It centers on a deubiquitinase OTUD5, which the authors suggest regulates the ubiquitination of a transcriptional repressor TRIM25. This in turn control the expression of PML, among other genes, to control tumorigenesis. The manuscript was well organized, the data was clear, logical and well put together, and the findings were new and interesting. I am therefore enthusiastic about the eventual impact of the manuscript and supportive of its publication. Below are a few points and suggested experiments that I would hope the reviewers can first address prior to full acceptance of the paper.

Response: The authors greatly appreciate the referee's positive, constructive and detailed comments and suggestions on our manuscript. We have studied the referee's comments carefully and tried our best to revise our manuscript according to these comments. We hope that this revised manuscript is now suitable for publication.

Major points

1. In figure 3, the authors make the case that TRIM25 ubiquitination is controlled by OTUD5. However, in all of the panels in this experiment, the authors are not directly assessing the covalent modification of TRIM25 with ubiquitin, and instead, rely on TRIM25 pulldowns followed by ubiquitin immunoblots. Thus, it is unclear if the smeared band in the pulldown is in fact TRIM25, or some other co-precipitating protein. I understand that these are routinely done under near-denaturing conditions, but this cannot rule out this concern. Since this is a top-line conclusion of the manuscript, it must be addressed. I do not think all of the experiment needs to be redone, but just one or two, to show convincingly that it is in fact TRIM25 that is ubiquitin modified. I can imagine this being done in two ways. First, they could blot the TRIM25 pulldowns for

TRIM25 (or its tag) to show that it in fact smears into high-molecular weight bands. Alternatively, the authors can pulldown ubiquitin and show that TRIM25 comes with it.

1.Response: The authors greatly appreciate the reviewer for critical and constructive comments and suggestions to improve the quality of the manuscript. We confirmed that the ubiquitin smear was indeed ubiquitinated TRIM25 by assaying the elute with antibodies against Flag (Fig. 3F, Supplementary fig. 2) and HA (Fig. 5E).

2. The authors suggest that the data in Fig 2 shows that the increased growth of OTUD5 depleted cells is TRIM25 dependent. However, there data are inconclusive in this respect. The depletion of TRIM25 reduces proliferation in cells with OTUD5, and the relative decrease in proliferation between control and OTUD5-depleted cells that are depleted for TRIM25 appears almost the same. I have no issue with the result, but believe it is being over-interpreted in this case.

2. Response: We thank and agree with the referee's comment on our manuscript. We have tuned-down the conclusion in the revised manuscript. We changed the expression -"suggesting the cell proliferation promoting effects of OTUD5 knockdown was dependent on TRIM25" to "suggesting the cell proliferation promoting effects of OTUD5 knockdown was possibly dependent on TRIM25" (page 8).

3. The authors refer to OTUD5 as a tumor suppressor in several places. The bar for calling it that is, I think, quite high. Since there is no good evidence that TRIM25 is lost or mutated in tumors, and by itself, would cause tumors, I think it would be more accurate to say that it has tumor suppressive properties. Not everything that suppresses proliferation is a tumor suppressor.

Response: The authors greatly appreciate the referee's comment for improving the quality of our manuscript. We tuned down all the description in the manuscript.

(Page 5: "as a tumor suppressor" was deleted;

Page 15 23th: “To further validate that OTUD5 is a bona fide tumor suppressor” to “To further validate OTUD5 roles in tumor suppression”;

Page 16: “as a tumor suppressor” to “displayed strong tumor suppressive properties”;

Page 17, 10th: “functions as a tumor suppressor through” to “functions through”;

Page 17, 15th: “functions as a tumor suppressor” to “inhibits tumor progression”;

Page 18: “tumor suppressor” to “regulator of tumor progression”.)

4. Since the ubiquitination of trim25 appears to not be degradative it would be interesting to know what the chain linkage is, or to at least show that it is not K48 linked. This could be done using a K48R mutated version of ubiquitin.

Response: The authors greatly appreciate the reviewer for the constructive suggestions to improve the quality of the manuscript. We performed K48-UIM/K63-UIM IP assay with Hep3B cells, described in the method section. The results showed that OTUD5 depletion increased K63 ubiquitin chain of TRIM25 (Supplementary fig. 3).

5. The mechanism for how trim25 is regulated is unclear. Although not necessary to figure out for publication, I would suggest simply testing if it might affect TRIM25 nuclear localization or chromatin binding. These experiments could be done easily, using existing reagents, with simple fractionation techniques.

Response: The authors greatly appreciate the referee’s the constructive suggestions to improve the quality of the manuscript. We performed western blot assay with chromatin fraction. There was a little increase of TRIM25 level at chromatin upon OTUD5 depletion (Supplementary fig. 4C).

6. The authors say that they tested to see if trim25 regulates PML and that it doesn’t, but the data is not shown. This is actually a rather important finding and the negative data should be included.

Response: The authors greatly thank the referee's critical and constructive comments and suggestions to improve the quality of the manuscript. We think the results you mentioned is about that OTUD5 can't deubiquitinate PML in vitro, which has been add into supplementary figure 4B. Meanwhile, we performed PML ubiquitination assay, and found that co-overexpression with TRIM25 moderately increased ubiquitination level of PML (Supplementary fig. 4A).

7. In a couple of places, the authors performed chip experiments and used IgG IPs in untransfected cells as a control for an HA IP in a transfected cell population. These should actually be done using an HA IP in control transfected cells. See figure 5 in particular.

7. Response: The authors greatly appreciate the referee's the positive comments on our manuscript. It was a misapprehension. We used IgG IPs as control because the cells used in those ChIP assays were all transfected with HA-tagged expressing plasmids. We amend the description of the legend to avoid misunderstanding.

8. Figure 3B is not well controlled since phosphatase inhibitors are only used to control some of these conditions. Because these will block all dephosphorylation it makes the interpretation very difficult.

8. Response: The authors greatly appreciate the referee for critical suggestion to improve the quality of the manuscript. We are sorry for the confusion. We added PI only during the OTUD5 enzyme was purified (lane3, Fig. 3B). When the protein was eluted by Flag pepitide, we did not add PI. Moreover, for the *in vitro* deubiquitination assay, a small volume of enzyme was added into the mixture. Therefore, we believe that the results are meaningful.

Minor points

1. Would be interested to test OTUD5 overexpression in some of these assays.

1.Response: The authors greatly appreciate the referee's the constructive suggestions to improve the quality of the manuscript. We test the effect of OTUD5

overexpression on colony formation assay. We found that overexpression of OTUD5 but not OTUD5/C224S inhibited cell proliferation (Fig. 3H).

2. The inputs in Fig 2A don't make any sense. There shouldn't be three inputs for this experiment. The authors need to clarify and explain better how this is done.

2. Response: The authors greatly appreciate the referee's critical suggestion to improve the quality of the manuscript. In the figure 2A, the sample used for the WB was prepared for mas-spec, and we confirmed the presence of TRIM25 in the elute from OTUD5 complex. Moreover, the WB experiments of figure 2B and 2C were not suitable, we delete the first lanes of lower panels in the revised manuscript.

3. Fig 2G- need to show TRIM25 in the input of the IP to interpret the result that the UIM mediates binding partially.

3. Response: The authors greatly appreciate the referee's comment on our manuscript. We performed WB analysis on TRIM25 protein levels to confirm that UIM mediates binding partially (Fig. 2F).

4. The authors should be clearer that the phosphorylation site has already been shown to modulate OTUD5 activity.

4. Response: The authors greatly appreciate the referee's suggestion. We quote more studies about this point in our manuscript (Huang OW, et al. Nat Struct Mol Biol 2012;19(2):171-5; Renatus M, Farady CJ. Structure. 2012;20(4):570-1; Koerver L, et al. Cell Death Differ. 2016;23(12):2019-2030.)

5. Authors should indicate that OTUD5 is also called DUBA.

5. Response: The authors greatly appreciate the referee's suggestion. We add the description according to this suggestion (page 3: "also been called DUBA").

6. Need show evidence that the TRIM25 shRNA works

6. Response: The authors greatly appreciate the referee's critical suggestion to improve the quality of the manuscript. We performed WB analysis to confirm the knockdown efficiency of TRIM25 (Fig. 6E).

7. Typo in fig 5H (mmRNA levels)

7. Response: The authors greatly appreciate the referee's critical comments to improve the quality of the manuscript. We correct it.

8. In figure 5 there is some confusion of the use of flag or ha tags between the text and figure

8. Response: The authors greatly appreciate the referee's critical comments to improve the quality of the manuscript. We have corrected it.

9. Remove 2F and 5E- they are not needed

9. Response: The authors greatly appreciate the referee's critical comments to improve the quality of the manuscript. We have remove.

Reviewers' comments:

Reviewer #1 (Remarks to the Author):

In general, the manuscript has been greatly improved following this round of revisions. Most of my points have been duly addressed. That said, the point 3.4 did not convey its desired message.

The major idea of the manuscript is that OTUD5 deubiquitinates TRIM25 and decreases proliferation and tumour growth.

My comment 3.4 was about generating TRIM25 K117R mutant (incapable of being self-ubiquitinated) cells or mouse model that would support the central idea of the manuscript. As it stands, I fully agree with Reviewer #2 who states that 'the role of TRIM25 deubiquitination in the hepatocarcinoma cell line xenografts is only correlative'.

Moreover, the lack of the above experiments highlights some problems with conclusions and data interpretation. In Discussion we can read that 'Fittingly, TRIM25 RING domain deletion mutant failed to promote cell proliferation because self-ubiquitination of TRIM25 was abolished.' Without, TRIM25 K117R mutant cells of mouse model one cannot make such claims as TRIM25 can influence cell biology by ubiquitinating many other targets.

My prediction is that TRIM25 K117R mutant cells or mouse model will lack pathological phenotype.

Until I am proven wrong, authors' major idea will only be speculative and should be presented this way.

Reviewer #2 (Remarks to the Author):

In the revised version of their manuscript, F. Li et al. present substantial additional data that strongly support the proposal: OTUD5 DUB activity contributes to tumor suppression in vivo through the negative regulation of TRIM25 and subsequent increase in the PML tumor suppressor. The authors improved the quality of the manuscript with new conclusive experiments. In addition to mechanistic insights involving OTUD5-mediated deubiquitination in hepatocarcinoma cell line growth, the authors have provided important new experiments performed in xenografted mice showing crosstalks between OTUD5, TRIM25 and PML. They generalize OTUD5 involvement in other solid cancers than hepatocarcinoma, notably with longer overall survival of patients with lung expressing high level of OTUD5.

Thus, the authors provided clear responses to the points raised from the initial manuscript and I support its publication in Nature Communication after minor modifications, listed below.

- Contribution of OTUD5-mediated deubiquitination of the TRIM25 targets within the OTUD5/TRIM25 complex, in addition to TRIM25 deubiquitination itself, is still not clear. The authors did not use denaturing condition to purify ubiquitinated TRIM25, but did elution using FLAG or HA peptides that would release TRIM25-associated proteins as well. The results performed on the two TRIM25 targets (supplementary fig5) are too weak to conclude in a way or another. Thus, as proposed previously, the authors should have tested effect of OTUD5/TRIM25 on global ubiquitination: this could be easily done using the samples presented in supplementary fig.4 or 5a for PML and MAVS ubiquitination. The His-tagged ubiquitin and denaturing pulldowns may be used for blotting with an anti-ubiquitin antibody (as FK2 antibody).

- More importantly, all ubiquitination assays have been performed from 293T cells. Assessing TRIM25 ubiquitination regarding OTUD5 expression in Hep3B or huh7 cells would be required to definitively link the biochemical mechanism to OTUD5/TRIM25 effects on liver cancer cell line growth and tumorigenesis.

- It is not completely clear whether the blots presented in fig.2B,C,F-bottom are inputs (after centrifugation?) or WCL?
- Some full blots should be shown as supplementary figures as TRIM25, OTUD5 (fig 3A,E) and PML (fig4.E)
- Legends should be carefully revised. As an example: legend of fig4M and 4N is lacking and 4I,J,K are not the good ones.
- "apoptosis" should be replaced by "cell survival" line 286 and in figure 4J.
- Some key references are missing in introduction and discussion:
Papers from R.H. Chen's lab described PML down-regulation in cancers (line 115-117) through another ubiquitin-ligase:
1Yuan, W.C. et al. A Cullin3-KLHL20 Ubiquitin ligase-dependent pathway targets PML to potentiate HIF-1 signaling and prostate cancer progression. *Cancer cell* 20, 214-228 (2011).
PML nuclear bodies are now considered as spherical structures more than doughnut shaped (line105). Two recent reviews in addition to the cited papers (2007, 2006) highlight new important findings on formation/functions of PML NBs notably in cancer, with anti- and pro-tumoral aspects:
2Hsu, K.S. & Kao, H.Y. PML: Regulation and multifaceted function beyond tumor suppression. *Cell Biosci* 8, 5 (2018).
3Lallemand-Breitenbach, V. & de The, H. PML nuclear bodies: from architecture to function. *Curr Opin Cell Biol* 52, 154-161 (2018).
The authors mention the anti-tumoral PML/p53 axis found in cell lines, they might add it is relevant in vivo, in cancer mouse models:
4Ablain, J. et al. Activation of a promyelocytic leukemia-tumor protein 53 axis underlies acute promyelocytic leukemia cure. *Nature medicine* 20, 167-174 (2014).

Reviewer #3 (Remarks to the Author):

The authors have addressed essentially all of the comments raised in my previous review. I think this is an interesting new study that adds to our understanding of how DUBs might promote cancer and am therefore fully supportive of its publication.

Reviewers' comments:

Reviewer #1 (Remarks to the Author):

In general, the manuscript has been greatly improved following this round of revisions. Most of my points have been duly addressed. That said, the point 3.4 did not convey its desired message. The major idea of the manuscript is that OTUD5 deubiquitinates TRIM25 and decreases proliferation and tumour growth. My comment 3.4 was about generating TRIM25 K117R mutant (incapable of being self-ubiquitinated) cells or mouse model that would support the central idea of the manuscript.

As it stands, I fully agree with Reviewer #2 who states that 'the role of TRIM25 deubiquitination in the hepatocarcinoma cell line xenografts is only correlative'. Moreover, the lack of the above experiments highlights some problems with conclusions and data interpretation. In Discussion we can read that 'Fittingly, TRIM25 RING domain deletion mutant failed to promote cell proliferation because self-ubiquitination of TRIM25 was abolished.' Without, TRIM25 K117R mutant cells of mouse model one cannot make such claims as TRIM25 can influence cell biology by ubiquitinating many other targets. My prediction is that TRIM25 K117R mutant cells or mouse model will lack pathological phenotype. Until I am proven wrong, authors' major idea will only be speculative and should be presented this way.

Response: The authors greatly appreciate the referee's constructive comment on our manuscript. This comment is very important for improving the quality of our manuscript.

We designed a colony formation assay to compare the effect of TRIM25-KR mutant and OTUD5 on tumor cell proliferation. The results demonstrated that TRIM25-KR mutant inhibited cell growth, consistently, the colonies formed by cells with increased OTUD5 expression was significantly less than that formed by the control cells, implying that the autoubiquitination of TRIM25 was crucial for its function in cell proliferation regulation (supplementary fig. 2C-D).

We revised and amended the discussion as "RING domain deletion and K117R mutant of TRIM25 substantially inhibited its function, implying that OTUD5 functions by regulating ubiquitination of TRIM25."

In this revised manuscript, we have incorporated additional experimental evidence to address the concern. We hope that this revised manuscript is now suitable for publication.

Reviewer #2 (Remarks to the Author):

In the revised version of their manuscript, F. Li et al. present substantial additional data that strongly support the proposal: OTUD5 DUB activity contributes to tumor suppression in vivo through the negative regulation of TRIM25 and subsequent increase in the PML tumor suppressor. The authors improved the quality of the manuscript with new conclusive experiments. In addition to mechanistic insights involving OTUD5-mediated deubiquitination in hepatocarcinoma cell line growth, the authors have provided important new experiments performed in xenografted mice showing crosstalks between OTUD5, TRIM25 and PML. They generalize OTUD5 involvement in other solid cancers than hepatocarcinoma, notably with longer overall survival of patients with lung expressing high level of OTUD5.

Thus, the authors provided clear responses to the points raised from the initial manuscript and I support its publication in Nature Communication after minor modifications, listed below.

Response: The authors greatly appreciate the referee's positive and detailed suggestions on our manuscript. We have studied the referee's comments carefully and tried our best to revise our manuscript. We hope that this revised manuscript is now suitable for publication.

1. Contribution of OTUD5-mediated deubiquitination of the TRIM25 targets within the OTUD5/TRIM25 complex, in addition to TRIM25 deubiquitination itself, is still not clear. The authors did not use denaturing condition to purify ubiquitinated TRIM25, but did elution using FLAG or HA peptides that would release TRIM25-associated proteins as well. The results performed on the two TRIM25 targets (supplementary fig5) are too weak to conclude in a way or another. Thus, as proposed previously, the authors should have tested effect of OTUD5/TRIM25 on global ubiquitination: this could be easily done using the samples presented in supplementary fig.4 or 5a for PML and MAVS ubiquitination. The His-tagged ubiquitin and denaturing pulldowns may be used for blotting with an anti-ubiquitin antibody (as FK2 antibody).

1. Response: The authors greatly appreciate the reviewer for critical and constructive suggestion to improve the quality of the manuscript. We detected

the denaturing pulldowns with anti-ubiquitin antibody using the samples in supplementary fig. 4a, and added the blotting images into the supplementary figures (supplementary fig. 4a). The results showed that the level of total ubiquitination was kept unchanged after 293T cells were transiently transfected with expression constructs encoding Myc-PML, HA-TRIM25 and His-ub.

2. More importantly, all ubiquitination assays have been performed from 293T cells. Assessing TRIM25 ubiquitination regarding OTUD5 expression in Hep3B or huh7 cells would be required to definitively link the biochemical mechanism to OTUD5/TRIM25 effects on liver cancer cell line growth and tumorigenesis.

2. Response: The authors greatly appreciate the referee's the positive comments on our manuscript. We had assayed the effect of OTUD5 expression on TRIM25 ubiquitination in Hep3B cell in Figure 3D and 3E. We detailed the description of the legend.

3. It is not completely clear whether the blots presented in fig.2B,C,F-bottom are inputs (after centrifugation?) or WCL?

3. Response: The authors greatly appreciate the reviewer for the constructive suggestions to improve the quality of the manuscript. We presented the results not properly and clearly in Figure 2B and 2C. We have revised Figure 2B and C and deleted redundant bottom lanes in both figures. We performed immunoprecipitation (IP) to detect endogenous OTUD5 and TRIM25 interaction with the same cell lysate by control IgG or OTUD5 antibody (Figure 2B), and reversely, control IgG or TRIM25 antibody (Figure 2C). The first lane is input, second lane is IP from control IgG, and third lane is IP from OTUD5 or TRIM25 antibody in Figure 2B and 2C.

In Figure 2F, the samples were inputs which were prepared from the cells used for immunoprecipitation assays. We labeled again and amended the legend accordingly.

4. Some full blots should be shown as supplementary figures as TRIM25, OTUD5 (fig 3A,E) and PML (fig4.E)

4. Response: The authors greatly appreciate the referee for the comments to improve the quality of the manuscript. We have put the full blots to the “source data”.

5. Legends should be carefully revised. As an example: legend of fig4M and 4N is lacking and 4I,J,K are not the good ones.

5. Response: The authors greatly appreciate the referee’s critical comments. We carefully revised the legends to avoid such mistakes.

6. “apoptosis” should be replaced by “cell survival” line 286 and in figure 4J.

6. Response: The authors greatly appreciate the referee’s comment to improve the manuscript. We amend the description according to the suggestion.

7. Some key references are missing in introduction and discussion:

Papers from R.H. Chen’s lab described PML down-regulation in cancers (line 115-117) through another ubiquitin-ligase:

1Yuan, W.C. et al. A Cullin3-KLHL20 Ubiquitin ligase-dependent pathway targets PML to potentiate HIF-1 signaling and prostate cancer progression. *Cancer cell* 20, 214-228 (2011).

PML nuclear bodies are now considered as spherical structures more than doughnut shaped. Two recent reviews in addition to the cited papers (2007, 2006) highlight new important findings on formation/functions of PML NBs notably in cancer, with anti- and pro-tumoral aspects:

2Hsu, K.S. & Kao, H.Y. PML: Regulation and multifaceted function beyond tumor suppression. *Cell Biosci* 8, 5 (2018).

3Lallemand-Breitenbach, V. & de The, H. PML nuclear bodies: from architecture to function. *Curr Opin Cell Biol* 52, 154-161 (2018).

The authors mention the anti-tumoral PML/p53 axis found in cell lines, they might add it is relevant in vivo, in cancer mouse models:

4Ablain, J. et al. Activation of a promyelocytic leukemia-tumor protein 53 axis underlies acute promyelocytic leukemia cure. *Nature medicine* 20, 167-174 (2014).

7. Response: The authors greatly appreciate the referee's suggestion to improve the quality of the manuscript. Indeed, the studies listed should be mentioned in our manuscript, and we have quoted these studies in the proper position in our manuscript, and in the references.

(references 42--Yuan, W.C. et al. A Cullin3-KLHL20 Ubiquitin ligase-dependent pathway targets PML to potentiate HIF-1 signaling and prostate cancer progression. *Cancer cell* 20, 214-228 (2011).

references 33--Hsu, K.S. & Kao, H.Y. PML: Regulation and multifaceted function beyond tumor suppression. *Cell Biosci* 8, 5 (2018).

references 34--Lallemand-Breitenbach, V. & de The, H. PML nuclear bodies: from architecture to function. *Curr Opin Cell Biol* 52, 154-161 (2018).

The authors mention the anti-tumoral PML/p53 axis found in cell lines, they might add it is relevant in vivo, in cancer mouse models:

references 36--Ablain, J. et al. Activation of a promyelocytic leukemia-tumor protein 53 axis underlies acute promyelocytic leukemia cure. *Nature medicine* 20, 167-174 (2014).)

Reviewer #3 (Remarks to the Author):

The authors have addressed essentially all of the comments raised in my previous review. I think this is an interesting new study that adds to our understanding of how DUBs might promote cancer and am therefore fully supportive of its publication.

Response: The authors appreciate the referee 's kindly encourage and hard work.

REVIEWERS' COMMENTS:

Reviewer #1 (Remarks to the Author):

The manuscript is ready for publication.

Reviewer #2 (Remarks to the Author):

The manuscript has been improved again and the authors have provided appropriate answers to all the points I raised and they have provided relevant revisions.

Some style issues might be still corrected, as in the abstract, line 31-32:

"the growth-promoting effect of OTUD5 depletion was dependent on its interaction with TRIM25."
"its interaction with" should be removed, or the sentence rephrased.

All together, according to me, the manuscript is now suited for publication by Nature Communication.